# Intracerebroventricular insulin injection acutely normalizes the augmented exercise pressor reflex in male rats with type 2 diabetes mellitus

Juan A. Estrada[1] , Rie Ishizawa[2], Han-Kyul Kim[1,3] , Ayumi Fukazawa[1] , Amane Hori[1], Norio Hotta[4] , Gary A. Iwamoto[5], Scott A. Smith[1,3] , Wanpen Vongpatanasin[3] and Masaki Mizuno[1,3]

[1]*Department of Applied Clinical Research, University of Texas Southwestern Medical Center, Dallas, TX, USA*
[2]*National Institute of Fitness and Sports, Kanoya University, Kanoya, Japan*
[3]*Department of Internal Medicine, University of Texas Southwestern Medical Center, Dallas, TX, USA*
[4]*College of Life and Health Sciences, Chubu University, Kasugai, Japan*
[5]*Department of Surgery, University of Texas Southwestern Medical Center, Dallas, TX, USA*

Handling Editors: Vaughan Macefield & Marc Kaufman

The peer review history is available in the Supporting Information section of this article (https://doi.org/10.1113/JP286715#support-information-section).

**Juan Estrada** is an Instructor in the Department of Applied Clinical Research at UT Southwestern Medical Center in Dallas, TX. He received his bachelor's degree in Exercise Science from the University of Texas at Arlington, and his PhD degree in Biomedical Science from the University of North Texas Health Science Center in Fort Worth, TX. His research focuses on neural control of cardiovascular responses to exercise in health and disease.

The Journal of Physiology

**Abstract**   The exercise pressor reflex (EPR) is exaggerated in type 2 diabetes mellitus (T2DM), but the underlying central nervous system aberrations have not been fully delineated. Stimulation of muscle afferents within working skeletal muscle activates the EPR, by sending information to neurons in the brainstem, where it is integrated and results in reflexively increased mean arterial pressure (MAP) and sympathetic nerve activity. Brain insulin is known to regulate neural activity within the brainstem. We hypothesize that brain insulin injection in T2DM rats attenuates the augmented EPR, and that T2DM is associated with decreased brain insulin. Using male Sprague–Dawley rats, T2DM and control rats were generated via an induction protocol with two low doses of streptozotocin (35 and 25 mg/kg, I.P.) in combination with a 14–23-week high-fat diet or saline injections and a low-fat diet, respectively. After decerebration, MAP and renal sympathetic nerve activity (RSNA) were evaluated during EPR stimulation, evoked by electrically induced muscle contraction via ventral root stimulation, before and after (1 and 2 h post) intracerebroventricular (I.C.V.) insulin microinjections (500 mU, 50 nl). I.C.V. insulin decreased peak MAP ($\Delta$MAP Pre (36 $\pm$ 14 mmHg) *vs.* 1 h (21 $\pm$ 14 mmHg) *vs.* 2 h (11 $\pm$ 6 mmHg), $P < 0.05$) and RSNA ($\Delta$RSNA Pre (107.5 $\pm$ 40%), *vs.* 1 h (75.4 $\pm$ 46%) *vs.* 2 h (51 $\pm$ 35%), $P < 0.05$) responses in T2DM, but not controls. In T2DM rats, cerebrospinal fluid insulin was decreased (0.41 $\pm$ 0.19 *vs.* 0.11 $\pm$ 0.05 ng/ml, control ($n = 14$) *vs.* T2DM ($n = 4$), $P < 0.01$). The results demonstrated that insulin injections into the brain normalized the augmented EPR in brain hypoinsulinaemic T2DM rats, indicating that the EPR can be regulated by brain insulin.

(Received 9 April 2024; accepted after revision 15 July 2024; first published online 20 August 2024)

**Corresponding author** M. Mizuno: Department of Applied Clinical Research, School of Health Professions, University of Texas Southwestern Medical Centre, 5323 Harry Hines Boulevard, Dallas, TX 75390-9174, USA. Email: masaki.mizuno@utsouthwestern.edu

**Abstract figure legend** The exercise pressor reflex (EPR), evoked by muscle contraction, is exaggerated in male rats with type 2 diabetes mellitus (T2DM). Cerebrospinal fluid insulin and phosphoinositide 3-kinase in the nucleus tractus solitarius levels are normal in controls and low in T2DM rats, corresponding with normal and exaggerated EPR function. Injection of exogenous insulin into the brain normalizes the EPR only in T2DM rats. Created with BioRender.com.

## Key points

- The reflexive increase in blood pressure and sympathetic nerve activity mediated by the autonomic nervous system during muscle contractions is also known as the exercise pressor reflex.
- The exercise pressor reflex is dangerously augmented in type 2 diabetes, in both rats and humans.
- In type 2 diabetic rats both cerebrospinal fluid insulin and phosphoinositide 3-kinase signalling within cardiovascular brainstem neurons decrease in parallel.
- Brain insulin injections decrease the magnitude of the reflexive pressor and sympathetic responses to hindlimb muscle contraction in type 2 diabetic rats.
- Partial correction of low insulin within the central nervous system in type 2 diabetes may treat aberrant exercise pressor reflex function.

## Introduction

During physical activity, the cardiovascular system is regulated in part by the exercise pressor reflex (EPR). The EPR is activated via afferent feedback from muscle sensory fibres (McCloskey & Mitchell, 1972). The muscle afferents are stimulated by mechanical (Hayes & Kaufman, 2001; Hayes et al., 2009; Kaufman et al., 1983) and metabolic/chemical stimuli associated with skeletal muscle contraction (Kaufman et al., 1983, 1984). These muscle afferents project onto neurons in the cardiovascular control centres of the hindbrain, including the NTS (Craig, 1995; Potts et al., 2002). Activation of the EPR results in increases in blood pressure, heart rate (HR) and sympathetic outflow during exercise to increase blood flow to working skeletal muscle, to match its requirements for oxygen and metabolic activity (Grotle et al., 2020). As recently reviewed, in humans with type 2 diabetes

mellitus (T2DM) the EPR is detrimentally augmented, or exaggerated, in adolescents, middle-aged adults and older adults (Grotle & Stone, 2019). However, our current understanding of the pathophysiology in T2DM with respect to the EPR in humans remains limited. A more elaborate delineation of the mechanisms underlying these pathological changes would improve our understanding of the relationship between the EPR and T2DM. Furthermore, this knowledge may aid in improving the sustainability of exercise recommendations, since exercise is a critical tool used in the management of T2DM (Kirwan et al., 2017).

We have previously demonstrated that the EPR is augmented in a rat model of T2DM (Kim et al., 2019). This animal model of chemically and dietary-induced T2DM produces abnormal increases in the cardioaccelerator, pressor and sympatho-excitatory responses to hindlimb muscle contraction, characteristic of the observations in diabetic middle-aged humans (Holwerda et al., 2016). The augmented EPR responses in this animal model can in part be explained by increased muscle afferent sensitivity (Ishizawa et al., 2021). Various circulating factors such as glucose or insulin can sensitize both the peripheral chemo- (Hori et al., 2022; Ishizawa et al., 2021) and mechano-sensitive afferents (Hotta et al., 2019). On the other hand, much less is known about the central neuronal alterations of the EPR pathway in T2DM, although insulin has important peripheral and central actions in this disease (Agrawal et al., 2021). Interestingly, a reduced brain insulin transport was reported in obese hyperinsulinaemic Zucker rats (Schwartz et al., 1990). So, central hypoinsulinaemia in T2DM may be implicated in abnormal reflex control of the circulation. In support of this concept, earlier studies have suggested that insulin receptor (IR) signalling can hyperpolarize neurons through the phosphoinositide 3-kinase (PI3K) pathway, which activates $K_{ATP}$ channels and causes decreased neuronal excitability (Plum et al., 2006). Thus, it is a logical proposition that insulin signalling modulates the excitability of neurons within the cardiovascular control centres of the hindbrain during EPR activation (Mizuno et al., 2021).

In this study, therefore, it was hypothesized that increased neuronal excitability during central hypo-insulinaemia is one of the mechanisms underlying the exaggerated EPR in T2DM. The primary aim of the current investigation was to determine whether intra-cerebroventricular injection of insulin in T2DM rats would attenuate the augmented EPR response to hindlimb muscle contraction. Our secondary aim was to determine whether T2DM is associated with central hypo-insulinaemia, and whether concomitant alterations in the insulin signalling pathway are present in these rats. That is, if the expression and activity of IR and its major downstream signalling enzymes, PI3K and Akt, is decreased in

T2DM rats within the NTS, a major cardiovascular control region populated with IR-positive neurons (Estrada et al., 2023).

## Methods

All the experiments described were performed in accordance with the US Department of Health and Human Services NIH *Guide for the Care and Use of Laboratory Animals*. All surgical and experimental procedures were approved by the Institutional Animal Care and Use Committee of the University of Texas Southwestern Medical Center (no.2019-102849). All authors understood and the study conformed to the guidelines and ethical principles of *The Journal of Physiology* (Grundy, 2015).

### Streptozotocin injections and dietary protocols

To generate the T2DM phenotype in male Sprague–Dawley rats (Inotiv, Indianapolis, IN, USA) for physiological and biochemical studies, two low-dose streptozotocin (STZ) injections were administered via the intraperitoneal (I.P.) route, followed by high-fat diet (HFD) feeding, as described previously (Kim et al., 2019). Briefly, STZ (Sigma-Aldrich, St Louis, MO, USA), was administered twice in young adult male rats (7–8 weeks of age). The first STZ injection was given at a dose of 35 mg/kg (I.P.), then 7 days later the second STZ injection was given at a dose of 25 mg/kg. In parallel, a separate group of rats were treated with saline in place of STZ as the vehicle control. A week after the final injection, the STZ treated rats were fed a HFD (42.0% fat, 42.8% carbohydrate, and 15.2% protein, TD96121, Harlan Teklad, Madison, WI, USA, *n = 13*). The saline treated rats were fed a low-fat diet (LFD) (normal chow diet: 13.0% fat, 67.9% carbohydrate, and 19.1% protein, TD08485, Harlan Teklad, *n = 14*). The feeding window of the STZ/HFD fed rats was 102−164 days (∼15–23 weeks of age), while the feeding window of LFD fed rats was 97−156 days (∼14–22 weeks of age). All rats were housed in temperature (22−24°C) and humidity (40−60%) controlled chambers (EcoFlo, Allentown, PA, USA), one to four per cage, under a 12 h light/dark cycle and given *ad libitum* access to food and water for the duration of the study. On the day of terminal experiments all rats were subjected to an overnight fast.

### General surgical procedures for intracerebroventricular microinjection studies

All rats used in the physiological experiments described below were rendered insentient by mechanical decerebration, and then subjected to

intracerebroventricular (ɪ.c.v.) microinjections. The general surgical procedures used in these studies have been described elsewhere (Mizuno et al., 2011; Smith et al., 2001). In brief, anaesthesia induction (4% iso-flurane, oxygen balanced) was first carried out using a small chamber connected to a gas vaporizer. Once adequate anaesthetic depth was confirmed by absence of the tail and toe-pinch responses, the rat was moved onto a surgical table equipped with a far-infrared warming pad (RightTemp; Kent Scientific, Torrington, CT, USA) and a nosecone connected to the anaesthesia vaporizer. The body temperature was kept within a constant range of 36.5–37.5°C throughout surgical and experimental protocols, and anaesthesia was maintained at 2% isoflurane throughout surgical protocols until after decerebration. A tracheostomy was then performed, and the trachea was intubated for mechanical ventilation (Model 683; Harvard Apparatus, Holliston, MA, USA) under constant anaesthesia (2% isoflurane). A retrograde catheter (RPT-40; Braintree Scientific, Braintree, MA, USA) was then inserted into the right carotid artery until the tip of the catheter reached the bifurcation point with the aorta, to measure arterial blood pressure (ABP). The arterial catheter was connected to a pressure transducer (MLT0380/D; ADInstruments Inc., Colorado Springs, CO, USA). The left carotid artery was then ligated to prevent excessive bleeding during decerebration. The right jugular vein was then also fitted with a retrograde catheter to deliver a constant ɪ.v. infusion of fluids (3–5 ml/h/kg of a cocktail constituting 2 ml of 1 M $NaHCO_3$, 10 ml of 5% dextrose, and 38 ml of lactated Ringer's solution) to ensure adequate hydration, plasma volume and arterial pressure throughout surgical and experimental procedures. Needle electrodes were used to measure electrocardiogram signals. The R wave derived from the electrocardiogram was used to derive HR.

To perform hindlimb muscle contractions, the ventral roots were electrically stimulated at $2-3 \times$ motor threshold (MT). First, a laminectomy was performed at the $L_2-L_6$ vertebrae to gain access to the spinal cord. The dura mater was cut and reflected, the left $L_4$ and $L_5$ ventral roots were isolated, cut, and the spinal cavity was kept covered and moist with mineral oil and cotton. Contraction of the hindlimb muscles was confirmed by placing the cut ends of the roots onto a pair of bipolar stainless-steel electrodes and briefly applying an electrical stimulation using a stimulator (Grass S88; Grass Instruments, Warwick, RI, USA) and constant current stimulation apparatus (PSIU6; Grass Instruments) through the roots (40 Hz, 0.1 ms pulse duration, $3 \times$ MT). The left hindlimb was then connected to a force transducer (FT10; Grass Instruments) to measure muscle tension. To make the connection the calcaneal bone was first cut, and a braided fishing line was then fixed onto

the calcaneal tendon, and the opposite end of the line was secured onto the force transducer. To further immobilize the hindlimb during contractions, the pelvic and ankle bones were secured with clamps, and then fixed onto a custom-made stereotaxic frame (David Kopf Instruments, Tujunga, CA, USA). The baseline muscle tension was set at $50-100$ g before contractions were performed.

To record changes in renal sympathetic nerve activity (RSNA) during muscle contraction and EPR activation, a pair of platinum recording electrodes were used. Using a retroperitoneal approach, the left kidney was surgically exposed, and the cavity was then filled with mineral oil. The renal nerve was then isolated, and the electrodes (Platinum; AM Systems, Sequim, WA, USA) were placed underneath, then fixed and insulated onto the nerve using a silicone adhesive mould (Kwik-Cast; WPI, Sarasota, FL, USA). The cavity was then closed and sutured. The recording electrode was connected to a high impedence headstage (NeuroAmp EX Headstage; ADInstruments Inc.) and the electrical signal was then sent to a pre-amplifier (NeuroAmp EX; ADInstruments Inc.), where RSNA was quantified by running the signal through a band-pass filter set at $150-2000$ Hz, then full-wave rectified, and low-pass filtered with a cut-off frequency of 30 Hz.

A precollicular decerebration was then performed to render the rats insentient before experimental protocols. First, the head was fixed onto the stereotaxic frame, and dexamethasone (0.2 mg ɪ.v.) was given to attenuate brain oedema during mechanical decerebration procedures. Isoflurane was then reduced (1%) to allow blood pressure to elevate before beginning decerebration. A craniotomy was then performed above the sagittal sinus, followed by making a section <1 mm rostral to the superior colliculus, and finally aspirating all brain tissue rostral to the section. The cavity was then filled with haemostatic powder (Surgicel; Ethicon, Raritan, NJ, USA) and cotton around the brainstem to minimize bleeding. Isoflurane anaesthesia was then discontinued, and mechanical ventilation was maintained with oxygen alone throughout the remainder of the experiment. To eliminate the depressive effects of residual anaesthesia on the pressor and sympathetic response to muscle contraction, a period of 1 h was allowed to elapse before beginning experimental protocols (Smith et al., 2001).

To perform ɪ.c.v. microinjections, a small hole was first drilled into the skull. As previously described (Pricher et al., 2008), the hole was drilled at 2 mm caudal to the interaural line, on the midline. The needle (32 gauge needle; WPI) was brought to a depth of 7.3 mm ventral to the surface of the skull. To inject insulin and vehicle solutions into the fourth ventricle we used a micro-syringe (Hamilton syringe; VWR, Missouri City, TX, USA) mounted on a micropump (UMP3; WPI) connected to a microsyringe controller (Micro4; WPI).

**Experimental protocols for I.C.V. microinjection experiments.** I.C.V. microinjection experiments were performed with insulin (Humulin-R; Eli Lilly and Company, Indianapolis, IN, USA) or artificial cerebrospinal fluid (aCSF; Harvard Apparatus) vehicle solutions. The effects of I.C.V. insulin treatment on haemodynamics (mean arterial pressure (MAP) and HR) and sympathetic activity (RSNA) were assessed at rest and during muscle contraction. MAP, HR, RSNA and muscle tension were measured for the 30 s period before, and for the 30 s period during hindlimb muscle contractions. The muscle contractions were performed before I.C.V. insulin (500 mU, 50 nl) injections, and then at 1 and 2 h after I.C.V. insulin injections. The insulin concentration was determined by an earlier study showing that I.C.V. insulin administration alters sympathetic nervous system via the PI3K pathway over a period of 3 h after the bolus injection (Rahmouni et al., 2004). In a separate group of saline/LFD and STZ/HFD fed rats, I.C.V. injections of aCSF were performed as a vehicle control experiment.

### End experiment and procedures

For all rats used in the physiological experiments described above, hexamethonium bromide (60 mg/kg) (Sigma Aldrich) was administered intravenously to verify that RSNA signal was recorded from post-ganglionic nerve activity. Background noise was subtracted from the RSNA data by measuring the signal 30 m after a bolus infusion of 4 M potassium chloride (2 ml/kg, I.V.).

### Fasting blood glucose and insulin studies

To determine whether changes in blood glucose or insulin occurred in STZ/HFD relative to controls, rats were fasted the night prior to terminal experiments. Following the overnight fast, the tail was gently heated with warm water, then the tail was clipped 2−3 cm from the tip and a blood sample was immediately assessed for glucose with a handheld glucose meter and strips (FreeStyle Precision Neo; Abbott, Chicago, IL, USA). A blood sample of 300 μl was then collected into tubes (Microvettes; Sarsdedt, Newton, NC, USA) to assess plasma insulin. The collected blood samples were centrifuged (2000 *g*, 20 m, at 4°C), and the plasma fraction was collected and stored at −80°C until further processing. In post-decerebrate rats blood glucose was also assessed before and following I.C.V. insulin injections.

Just prior to the decerebration procedure, CSF was collected from rats. First, the cisternae magna region was surgically exposed, then the dura was punctured using an allergy syringe (27G; BD, Franklin Lakes, NJ, USA), and a sample of CSF was aspirated. The head was temporarily tilted down at 15° to facilitate collection of the sample. The CSF samples were then stored at −80°C until assayed for insulin. Plasma and CSF insulin were assayed using a commercial kit (Ultra-Sensitive Rat Insulin ELISA Kit, cat. no. 90060; Crystal Chem, Elk Grove Village, IL, USA). We used several observations to confirm the T2DM phenotype. First, the presence of hyperglycaemia in STZ/HFD relative to control rats in the fasted conscious state as previously demonstrated in this model (Ishizawa et al., 2021; Kim et al., 2019). Second, elevated blood glucose in STZ/HFD relative to control rats 1 h following cessation of anaesthesia and decerebration procedures, as isoflurane exposure significantly augments blood glucose further in T2DM (Fang et al., 2020). Last, the presence of significant changes fasting plasma and CSF insulin in STZ/HFD relative to control rats as similarly demonstrated by others (Cummings et al., 2008; Huo et al., 2022).

### NTS micropunch sampling and western blot studies

To determine protein expression of insulin signalling proteins in the NTS of control and STZ/HFD rats, brain tissue was harvested from a separate batch of rats not used for *in vivo* physiological experiments. First, rats were decapitated under anaesthesia and the brains were then harvested and snap-frozen with liquid nitrogen, then stored at −80°C until further processing. Then, coronal brainstem sections were obtained using a stainless-steel brain matrix (0.5 mm coronal rat; Stoelting, Wood Dale, IL, USA) and carbon-steel blades (Feather Blades; Ted Pella Inc., Redding, CA, USA). The temperature of the brain and brain matrix was maintained at −10°C during slicing, and the brainstem slices were obtained from 1 mm rostral through 1 mm caudal to the calamus scriptorius. The slices of interest were then transferred onto frozen slides and placed onto a stainless-steel plate, kept cooled with ice and a layer of pulverized dry ice. The NTS was then visually identified with the aid of a microscope, and for each rat bilateral micropunches of the NTS were collected with a 1 mm punch (Precision Brain Punches; Ted Pella Inc.).

The NTS micropunches were then homogenized using an ultrasonicator with probe (Branson SFX 150 Sonifier; Fisher Scientific, Hampton, NJ, USA) and extraction buffer (RIPA; Thermo Fisher Scientific, Waltham, MA, USA), with added protease (Protease Inhibitor Cocktail; Sigma-Aldrich) and phosphatase inhibitors (PhosSTOP; Roche, Basel, Switzerland). Immediately following homogenization, the sample was centrifuged (17,000 *g*, 30 min, at 4°C) and the supernatant was stored at −80°C. Protein concentration in lysates was determined using a protein assay (BCA; Thermo Fisher Scientific). Following determination of protein concentrations, the samples were then loaded onto precast polyacrylamide gels for SDS-PAGE, followed by western blot detection for proteins of interest. Briefly, 5−20 μg of each sample

was loaded into each well (TGX; Bio-Rad Laboratories, Hercules, CA, USA), and the samples were then run through the gel with running buffer (Running Buffer; Bio-Rad). Then, proteins were transferred from gels onto polyvinylidene difluoride (PVDF) membranes (Immobilon-FL; EMD Millipore, Burlington, MA, USA) with transfer buffer (Flash Blot; Advansta, San Jose, CA, USA). Immediately following a 20 min transfer, the membranes were rinsed with water and then dried at 37°C for 15 min before beginning western blot procedures. A prestained molecular weight ladder (Chameleon Duo Prestained Protein Ladder; LI-COR Biosciences, Lincoln, NE, USA) was used to verify the resolution of proteins during electrophoresis and successful transfer of proteins onto PVDF membranes.

For western blot detection, membranes were first blocked with blocking buffer (Intercept Blocking Buffer; LI-COR Biosciences, or Bullet Blocking One Buffer; Nacalai USA, San Diego, CA, USA) and then probed overnight at 4°C with primary antibodies against proteins of interest (rabbit anti-phospho-Akt (Ser473) monoclonal antibody, 1:1000, cat. no. 4060, RRID#AB_2315049, Cell Signaling Technology (CST), Danvers, MA, USA; mouse anti-pan-Akt monoclonal antibody, 1:1000, cat. no. 2920, RRID#AB_1147620, CST; rabbit anti-phospho-PI3K polyclonal antibody, 1:1000, cat. no. 4228, RRID#AB_659940, CST; rabbit anti-PI3K polyclonal antibody, 1:1000, cat. no. 4292, RRID#AB_329869, CST; mouse anti-IR $\beta$ monoclonal antibody, 1:1000, cat. no. sc-57342, RRID#AB_784102, Santa Cruz Biotechnology (SCBT), Dallas, TX, USA; mouse anti-$\beta$-actin monoclonal antibody, 1:1000, cat. no. sc-47778, RRID#AB_2714189, SCBT; rabbit recombinant anti-$\beta$-actin monoclonal antibody, 1:1000, cat. no. 81115-1-RR, RRID#AB_2923704, Proteintech (PTG), Rosemont, IL, USA). Membranes were then washed with tris-buffered saline–Tween 20 (TBS-T), followed by probing with secondary antibodies (horseradish peroxidase (HRP)-conjugated Affinipure goat anti-rabbit IgG, 1:1000, cat. no. SA00001-2, RRID#AB_2722564, PTG; HRP-conjugated Affinipure goat anti-mouse IgG, 1:1000, cat. no. SA00001-1, RRID#AB_10956166, PTG; IRDye 800CW goat anti-rabbit, 1:5000, cat. no. 926-32211, RRID#AB_621843, LI-COR Biosciences; IRDye 690RD goat anti-mouse, 1:5000, cat. no. 926-68071, RRID#AB_2722565, LI-COR Biosciences) for 1 h at 4°C. Membranes were then washed with TBS-T and imaged either via chemiluminescence (ECL; Thermo Fisher Scientific) or fluorescence detection using an imaging system (Odyssey Fc Imager; LI-COR Biosciences). The ECL method was used to detect the target protein of interest found in lower abundance relative to the loading control target, $\beta$-actin. Then the membrane was rinsed or stripped (NewBlot PVDF; LI-COR Biosciences), and the antibody incubation steps were repeated to probe for $\beta$-actin using the fluorescence detection method. All primary and secondary antibodies were diluted in TBS-T buffer.

## Data handling and statistical analyses

Hindlimb muscle contractions were performed for a period of 30 s following a baseline period of 30 s of preload tension (50–100 g). During each contraction, muscle tension, cardiovascular and sympathetic responses were measured. Data for MAP, HR, RSNA and muscle tension were obtained with LabChart 7 data acquisition software (ADInstruments) and the PowerLab analog-to-digital converter (PowerLab8/30; ADInstruments) at a 1 kHz sampling rate. For cardio-vascular, sympathetic and tension measurements, the baseline values were subtracted from the values measured during the hindlimb muscle contraction period. One second averages were measured, and the baseline sampling period was considered as 100% baseline RSNA, with stimulation-induced changes in RSNA expressed as a percentage of this baseline.

Only animals used in physiological experiments were used in the analysis of morphometric characteristics and baseline haemodynamics and sympathetic activity. Samples obtained from animals not used in physiological experiments were included in the analysis of fasting glucose, expressed as mg/dl, and insulin, expressed as ng/ml. Additionally, only animals used in physiological experiments were used for comparisons of blood glucose before and following I.C.V. microinjections. In the ELISA analysis of plasma insulin, values that fell below the detection range of the assay (0.1 ng/ml) were not used, and samples that had excessive haemolysis were also not used.

For western blot experiments densitometry data were expressed in arbitrary units as the ratio of target protein to loading control (within blot), or as the ratio of phospho-protein to total target protein (across blots). Each lane represents a tissue sample for one animal. For the loading control within blots, $\beta$-actin was used as a housekeeping protein. To calculate the loading corrected density of each lane, the raw target protein density was first divided by the raw density of $\beta$-actin in the same lane. Then, the normalized optical density was calculated by taking the loading corrected density of each lane and dividing this value by the average value of the loading corrected densities of the control samples, for each membrane. Additionally, total target protein was also used as an internal loading control across blots since we were limited by primary antibodies and detection method. That is, densitometry data for phosphorylated (phospho-) target protein in each lane was first corrected for total target protein in the corresponding lane on a separate

**Table 1. Morphometric characteristics, blood glucose, CSF insulin and baseline haemodynamics**

|  | Control | T2DM |
|---|---|---|
| *n* | 14 | 13 |
| Body weight (g) | 499 ± 37 | 484 ± 45 |
| Heart weight/body weight (mg/g) | 27.12 ± 0.16 | 26.56 ± 0.30 |
| Heart weight/tibial length (mg/mm) | 2.4 ± 2.9 | 2.4 ± 2.7 |
| Lung weight/body weight (mg/g) | 7.70 ± 2.08 | 7.66 ± 1.94 |
| Fasting blood glucose (mg/dl) | 101.4 ± 13.6 | 178.3 ± 107.2* |
| Fasting plasma insulin (ng/ml)[a] | 1.53 ± 0.40 (*n* = 13) | 0.91 ± 0.27**** (*n* = 17) |
| Fasting CSF insulin (ng/ml) | 0.41 ± 0.19 | 0.11 ± 0.05** (*n* = 4) |
| 1% Isoflurane anaesthesia | | |
| MAP (mmHg) | 107 ± 13 | 117 ± 21 |
| HR (bpm) | 347 ± 32 | 367 ± 26 |
| RSNA (signal:noise) | 5.5 ± 2.8 (*n* = 12) | 6.0 ± 2.6 |

Values are means ± SD; *n* is number of rats.
[a] Includes rats not used in physiological studies.
*$P < 0.05$ *versus* control, **$P < 0.01$ *versus* control, ****$P < 0.0001$ *versus* control, analysed using unpaired *t* test. CSF, cerebrospinal fluid; HR, heart rate; MAP, mean arterial pressure; RSNA, renal sympathetic nerve activity; T2D, type 2 diabetes mellitus.

blot, then normalized for optical density, as described above. Image Studio 5 software (LI-COR Biosciences) was used to both acquire and measure densitometry data from western blot experiments. Fluorescent molecular mass markers for each membrane were converted to greyscale images and then cut and overlayed on the corresponding chemiluminescence image. While each image faithfully represents the results produced from each primary antibody, the two lowest molecular mass markers could not be resolved on 10% gels, and therefore the lowest portion of the membrane was cropped out of representative images (below). Adobe Photoshop CS4 software was used to enhance image brightness and contrast, and Fiji (NIH, Bethesda, MD, USA) was used to crop, annotate and arrange each blot on a single figure.

Data for i.c.v. microinjection studies were analysed by two-way repeated measures (RM) ANOVA and included main effects for (1) group as a between-subject factor (control *vs.* T2DM), (2) time as a within-subject factor (Pre, 1, 2 h), and (3) group × time as an interaction. If significance was detected, a *post hoc* analysis was performed using the Holms-Šidák or Tukey's method. Fasting blood glucose, fasting plasma insulin and densitometry data were analysed using Student's unpaired *t* test. A *P*-value of <0.05 was defined as statistically significant. All data analysis was performed using GraphPad Prism9 software (GraphPad Software, Boston, MA, USA) and represented as individual data points and means ± SD.

## Results

### Phenotype of control (saline/LFD) *vs.* T2DM (STZ/HFD) rats

Morphometric characteristics and baseline haemodynamics and sympathetic activity before decerebration are summarized in Table 1. We found no differences in body weight or morphology between control and T2DM groups. Furthermore, no differences were observed in resting MAP, HR and RSNA in non-decerebrated, anaesthetized rats. Additionally, we found that fasting blood glucose was significantly elevated in T2DM relative to control rats ($P = 0.013$ for unpaired *t* test). This coincided with a significant decrease of both fasting plasma insulin ($P < 0.0001$ for unpaired *t* test) and CSF insulin (in control *vs.* in T2DM, $P = 0.008$ for unpaired *t* test) in T2DM relative to control rats. Last, blood glucose remained significantly elevated in T2DM relative to control rats after decerebration (see below).

### Influence of i.c.v. microinjections on resting haemodynamics, sympathetic activity and blood glucose

Resting haemodynamics, sympathetic nerve activity and blood glucose before and after i.c.v. microinjections in non-anaesthetized decerebrated control and T2DM rats are summarized in Table 2. In the i.c.v. insulin studies, we found that pre-microinjection resting MAP trended

**Table 2. Resting haemodynamics, sympathetic nerve activity and blood glucose pre- and post-microinjections**

| Group | i.c.v. insulin | | i.c.v. vehicle | |
| --- | --- | --- | --- | --- |
| | Control | T2DM | Control | T2DM |
| *n* | 7 | 10[a] | 7 | 4 |
| Resting MAP (mmHg) | | | | |
| Pre-microinjection | 77 ± 7 | 92 ± 14 | 74 ± 10 | 91 ± 25 |
| 1 h post-microinjection | 81 ± 19 | 85 ± 9 | 71 ± 12 | 93 ± 29 |
| 2 h post-microinjection | 80 ± 19 | 77 ± 10** | 71 ± 9 | 89 ± 22 |
| Resting HR (bpm) | | | | |
| Pre-microinjection | 409 ± 41 | 398 ± 42 | 371 ± 28 | 395 ± 35 |
| 1 h post-microinjection | 405 ± 31 | 389 ± 49 | 375 ± 27 | 400 ± 16 |
| 2 h post-microinjection | 408 ± 34 | 397 ± 45 | 390 ± 22 | 389 ± 30 |
| Resting RSNA (S:N) | | | | |
| Pre-microinjection | 5.2 ± 2.0 | 6.3 ± 3.0 | 4.2 ± 2.0 (*n* = 5) | 6.5 ± 2.2 (*n* = 3) |
| 1 h Post-microinjection | 4.2 ± 1.9 | 4.9 ± 2.9 | 4.1 ± 1.3 | 5.7 ± 2.6 |
| 2 h Post-microinjection | 4.5 ± 2.0 | 4.1 ± 3.7 | 4.7 ± 1.5 | 5.9 ± 3.2 |
| Blood glucose (mg/dl) | | | | |
| Pre-microinjection | 188 ± 39 (*n* = 5) | 294 ± 94[††] (*n* = 6) | 126 ± 38 | 291 ± 184 (*n* = 3) |
| 1 h post-microinjection | 114 ± 45 | 213 ± 78[†] | 118 ± 31 | 281 ± 192 |
| 2 h post-microinjection | 82 ± 33 | 171 ± 49[†] | 105 ± 55 | 248 ± 225 |

Values are means ± SD; *n* is number of rats.
[a] One T2DM rat was used for an experiment with both vehicle and insulin i.c.v. microinjections.
**$P < 0.01$ *vs*. Pre-microinjection, [†]$P < 0.05$ *vs*. Control group, [††]$P < 0.01$ *vs*. Control group, analysed using two-way RM ANOVA and Tukey's *post hoc* analysis. CSF, cerebrospinal fluid; HR, heart rate; i.c.v., intracerebroventricular; MAP, mean arterial pressure; RSNA, renal sympathetic nerve activity; T2D, type 2 diabetes mellitus.

towards being elevated in T2DM relative to controls (for ANOVA group × time interaction: $P = 0.047$, time: $P = 0.171$, group: $P = 0.360$). This trend was also observed in the i.c.v. vehicle studies (for ANOVA group × time interaction: $P = 0.742$, time: $P = 0.810$, group: $P = 0.085$). Furthermore, in the i.c.v. insulin studies resting MAP significantly decreased only in the T2DM rats at 2 h post-microinjection, but this observation was not demonstrated in the i.c.v. vehicle studies (Table 2). Additionally, in the i.c.v. insulin studies, blood glucose was significantly elevated in the T2DM rats *versus* the control rats before, and at 1 and 2 h post-microinjection (for ANOVA group × time interaction: $P = 0.913$, time: $P < 0.001$, group: $P = 0.010$). This trend was also observed in the i.c.v. vehicle studies (for ANOVA group × time interaction: $P = 0.638$, time: $P = 0.058$, group: $P = 0.063$). However, no differences in blood glucose were observed following i.c.v. microinjections of either insulin or vehicle solutions in control and T2DM rats,

although blood glucose trended downwards with time (Table 2). No significant differences were observed in resting HR between T2DM and control rats in either i.c.v. insulin (for ANOVA group × time interaction: $P = 0.963$, time: $P = 0.789$, group: $P = 0.474$) or vehicle studies (for ANOVA group × time interaction: $P = 0.344$, time: $P = 0.820$, group: $P = 0.213$). Likewise, no differences were observed in resting RSNA between T2DM and control rats in either i.c.v. insulin (for ANOVA group × time interaction: $P = 0.290$, time: $P = 0.013$, group: $P = 0.733$) or vehicle studies (for ANOVA group × time interaction: $P = 0.233$, time: $P = 0.362$, group: $P = 0.286$).

## Influence of brain insulin on the EPR in T2DM and control rats

The influence of brain insulin microinjections on the pressor and sympathetic responses to EPR activation are illustrated in Fig. 1. In control rats (Fig. 1*A*) the pressor

and sympathetic responses before insulin microinjections did not change at either 1 or 2 h after microinjections. Likewise, baseline arterial blood pressure and sympathetic activity also did not change before or after i.c.v. insulin microinjections. In contrast, in T2DM rats (Fig. 1*B*) the pressor and sympathetic responses before insulin microinjections are elevated compared to control rats, decreasing at 1 h, then further decreasing 2 h after insulin microinjections. In the T2DM rat baseline arterial blood pressure did not change following i.c.v. insulin micro-injection, with a slight decrease in baseline sympathetic activity.

In Fig. 2, analysis of the pressor and sympathetic responses to EPR activation before and after i.c.v. insulin microinjections demonstrates that the EPR is attenuated after insulin injections in the T2DM rats. The peak change in MAP responses (Fig. 2*A*) decreases significantly in T2DM rats following insulin microinjection (Pre: 19 ± 7 mmHg, 1 h: 17 ± 10 mmHg, 2 h: 11 ± 8 mmHg in control *vs*. Pre: 35 ± 13 mmHg, 1 h: 21 ± 14 mmHg, 2 h: 11 ± 6 mmHg in T2DM, $P = 0.008$ for ANOVA group × time interaction). Correspondingly, the integrated change in MAP (Fig. 2*B*) decreased in the T2DM rats (Pre: 98 ± 82 mmHg × s, 1 h: 203 ± 194 mmHg × s, 2 h: 167 ± 210 mmHg × s in control *vs*. Pre: 288 ± 302 mmHg × s, 1 h: 183 ± 194 mmHg × s, 2 h: 86 ± 94 mmHg in T2DM, $P = 0.046$ for ANOVA group × time interaction). Additionally, the peak change in RSNA responses (Fig. 2*E*) decreased significantly following insulin microinjections in the T2DM rats (Pre: 55 ± 20%, 1 h: 58 ± 44%, 2 h: 35 ± 16% in control *vs*. Pre: 108 ± 40%, 1 h: 58 ± 44%, 2 h: 35 ± 16% in T2DM, $P = 0.023$ for ANOVA group × time interaction). Correspondingly, the integrated change in RSNA (Fig. 2*F*) decreased in the T2DM rats (Pre: −2 ± 226% × s, 1 h: 90 ± 382% × s, 2 h: 82 ± 200% × s in control *vs*. Pre: 705 ± 517% × s, 1 h: 452 ± 292% × s, 2 h: 321 ± 273% × s in T2DM, $P = 0.049$ for ANOVA group × time interaction).

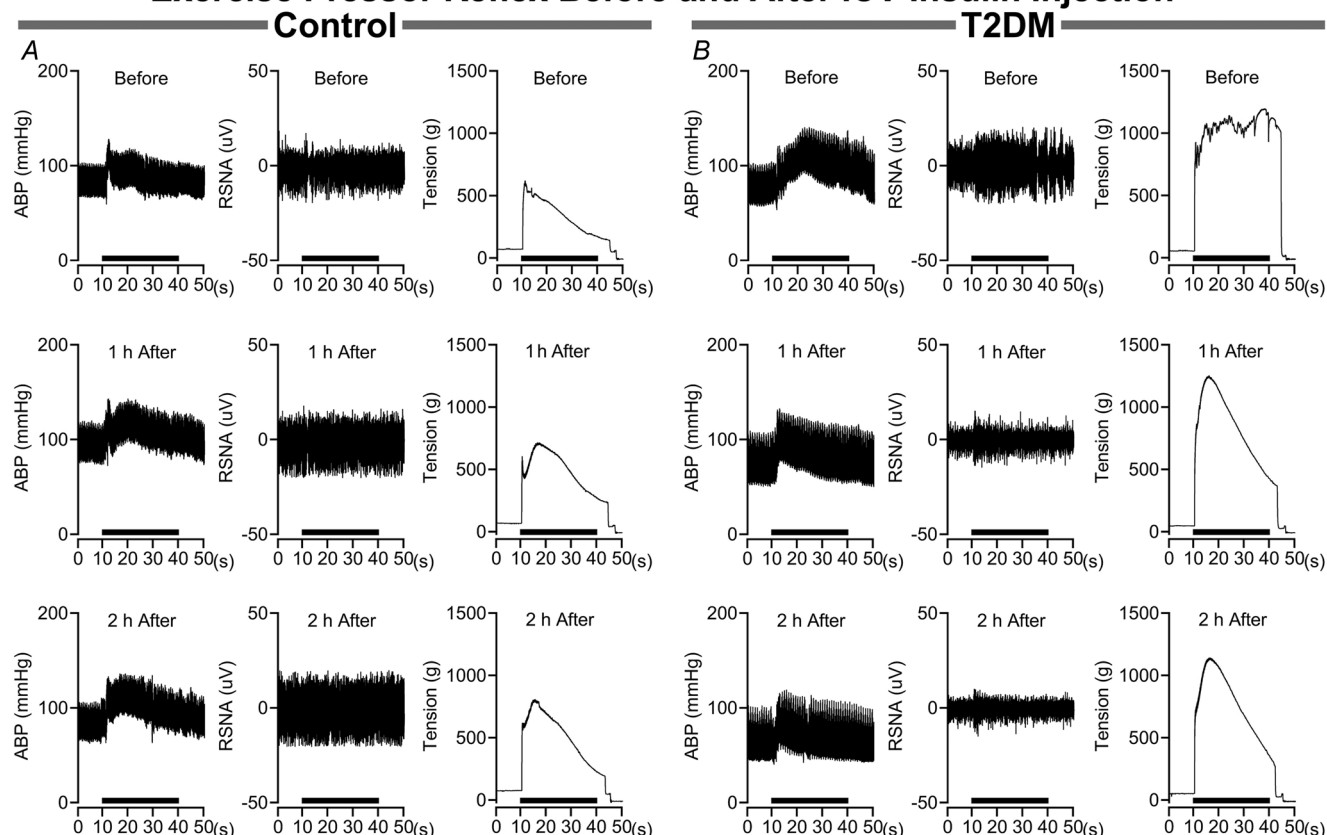

**Figure 1. Influence of intracerebroventricular insulin injections on the exercise pressor reflex in control and T2DM rats**
*A*, in control rats brain insulin treatment has no influence on the arterial blood pressure (ABP) and RSNA responses during the EPR stimulation period. *B*, in T2DM rats the ABP and RSNA responses during the EPR stimulation period decrease at 1 h after, then decreases further at 2 h after brain insulin treatment. The EPR stimulation period (30 s) is represented by the black bar above the *x*-axis (seconds). Data are raw blood pressure, RSNA and muscle tension tracings before and after i.c.v. insulin injections.

The peak change in HR (Fig. 2*C*) did not change following insulin microinjections in the T2DM rats (Pre: 12 ± 6 beats/min, 1 h: 11 ± 8 beats/min, 2 h: 10 ± 7 in control *vs.* Pre: 15 ± 8 beats/min, 1 h: 11 ± 7 beats/min, 2 h: 10 ± 7 in T2DM). However, the corresponding changes in the integrated HR response (Fig. 2*D*) decreased significantly (Pre: 118 ± 198 bpm × s, 1 h: 202 ± 179 beats/min × s, 2 h: 114 ± 101 bpm × s in control *vs.* Pre: 343 ± 242 bpm × s, 1 h: 207 ± 180 bpm × s, 2 h: 130 ± 115 bpm × s in T2DM, *P* = 0.039 for ANOVA group × time interaction). The peak tension during contractions (Fig. 2*G*) did not change following insulin microinjections in T2DM rats (Pre: 834 ± 298 g, 1 h: 701 ± 369 g, 2 h: 571 ± 352 g in control *vs.* Pre: 975 ± 292 g, 1 h: 960 ± 429 g, 2 h: 900 ± 424 g in T2DM). Correspondingly, the integrated tension (Fig. 2*H*) during contractions did not change (Pre: 15 927 ± 7781 g × s, 1 h: 14 688 ± 7540 g × s, 2 h: 11 445 ± 6893 g × s in control *vs.* Pre: 20 628 ± 6497 g × s, 1 h: 19 498 ± 7936 g × s, 2 h: 18 527 ± 8172 g × s for T2DM).

The influence of brain vehicle microinjections on the pressor and sympathetic responses to EPR activation are illustrated in Fig. 3. In control rats (Fig. 3*A*) the pressor and sympathetic responses before insulin microinjection did not change at either 1 or 2 h after microinjections. Likewise, baseline arterial blood pressure and sympathetic activity did not change following vehicle microinjections. In parallel, the pressor and sympathetic responses to EPR activation in T2DM rats (Fig. 3*B*) are not influenced by vehicle microinjections, nor were the corresponding baseline parameters.

In Fig. 4, analysis of the pressor and sympathetic responses to EPR activation before and after i.c.v. vehicle microinjections demonstrates that the EPR does not change in either the control or the T2DM rats. The peak change in MAP responses (Fig. 4*A*) did not change in either the control or the T2DM rats following vehicle microinjections (Pre: 20 ± 5 mmHg, 1 h: 23 ± 10 mmHg, 2 h: 21 ± 10 mmHg in control *vs.* Pre: 41 ± 14 mmHg, 1 h: 44 ± 15 mmHg, 2 h: 34 ± 14 mmHg in T2DM). Correspondingly, the integrated change in MAP (Fig. 4*B*) did not change (Pre: 257 ± 106 mmHg × s, 1 h: 342 ± 152 mmHg × s, 2 h: 343 ± 183 mmHg × s in control *vs.* Pre: 462 ± 172 mmHg × s, 1 h: 485 ± 159 mmHg

## ICV Insulin Injections

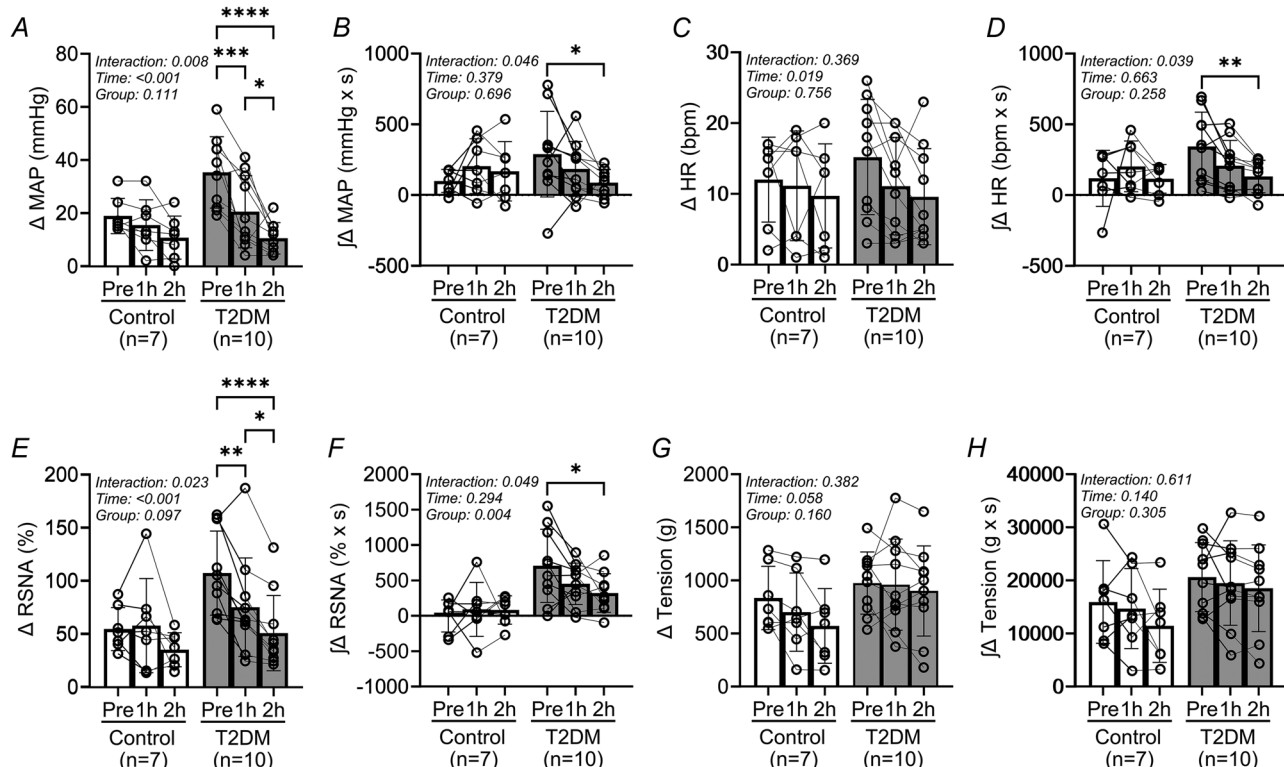

**Figure 2. Intracerebroventricular insulin injections attenuate the exercise pressor reflex in T2DM rats**
Vertical bars represent the mean pressor (*A* and *B*), cardioaccelerator (*C* and *D*), and sympathetic (*E* and *F*) responses, and hindlimb muscle tension (*G* and *H*) during EPR stimulation. These variables were measured before (Pre) and after (1 h and 2 h) i.c.v. insulin injections in control (open bars and circles) and T2DM (filled bars and circles) rats. Analysed by two-way RM ANOVA and Šidák's multiple comparison test. *$*P < 0.05$, **$P < 0.01$, ***$P < 0.001$, ****$P < 0.0001$. Bars are the mean ± SD.

× s, 2 h: 378 ± 254 mmHg in T2DM). Additionally, the peak change in RSNA responses (Fig. 4*E*) did not change in either the control or the T2DM rats following vehicle microinjections (Pre: 55 ± 19%, 1 h: 79 ± 31%, 2 h: 42 ± 18% in control *vs.* Pre: 151 ± 98%, 1 h: 162 ± 105%, 2 h: 168 ± 144% in T2DM). However, the corresponding integrated RSNA responses (Fig. 4*F*) increased in T2DM (Pre: 538 ± 132% × s, 1 h: 419 ± 256% × s, 2 h: 173 ± 334% × s in control *vs.* Pre: 582 ± 468% × s, 1 h: 1006 ± 614% × s, 2 h: 1107 ± 606% × s in T2DM, *P* = 0.002 for ANOVA group × time interaction). It should be noted that the number of successful RSNA recordings was limited in the i.c.v. vehicle experiments, which may account for the latter finding.

The peak change in HR (Fig. 4*C*) was not different following vehicle microinjections in either the control or the T2DM rats (Pre: 31 ± 9 beats/min, 1 h: 35 ± 20 beats/min, 2 h: 21 ± 8 in control *vs.* Pre: 26 ± 10 beats/min, 1 h: 34 ± 18 beats/min, 2 h: 27 ± 13 in T2DM). Correspondingly, the integrated HR responses (Fig. 4*D*) did not change (Pre: 634 ± 172 bpm × s, 1 h: 748 ± 416 bpm × s, 2 h: 439 ± 196 bpm × s in control *vs.* Pre: 617 ± 301 bpm × s, 1 h: 693 ± 448 bpm × s, 2 h: 483 ± 260 bpm × s in T2DM). The peak tension during contractions (Fig. 4*G*) was higher following vehicle microinjections in T2DM rats (Pre: 959 ± 268 g, 1 h: 880 ± 177 g, 2 h: 980 ± 237 g in control *vs.* Pre: 805 ± 99 g, 1 h: 1,177 ± 135 g, 2 h: 1,221 ± 75 g in T2DM, *P* = 0.010 for ANOVA group × time interaction). However, the corresponding integrated tension (Fig. 4*H*) did not change (Pre: 16 279 ± 6568 g × s, 1 h: 16 822 ± 4244 g × s, 2 h: 19 462 ± 5265 g × s in control *vs.* Pre: 17 095 ± 1790 g × s, 1 h: 25 760 ± 3610 g × s, 2 h: 25 435 ± 4164 g × s for T2DM) in either group.

## Impact of T2DM on insulin signalling in the NTS

Representative images of western blots are shown in Fig. 5. Proteins with a molecular mass lower than 25 kDa

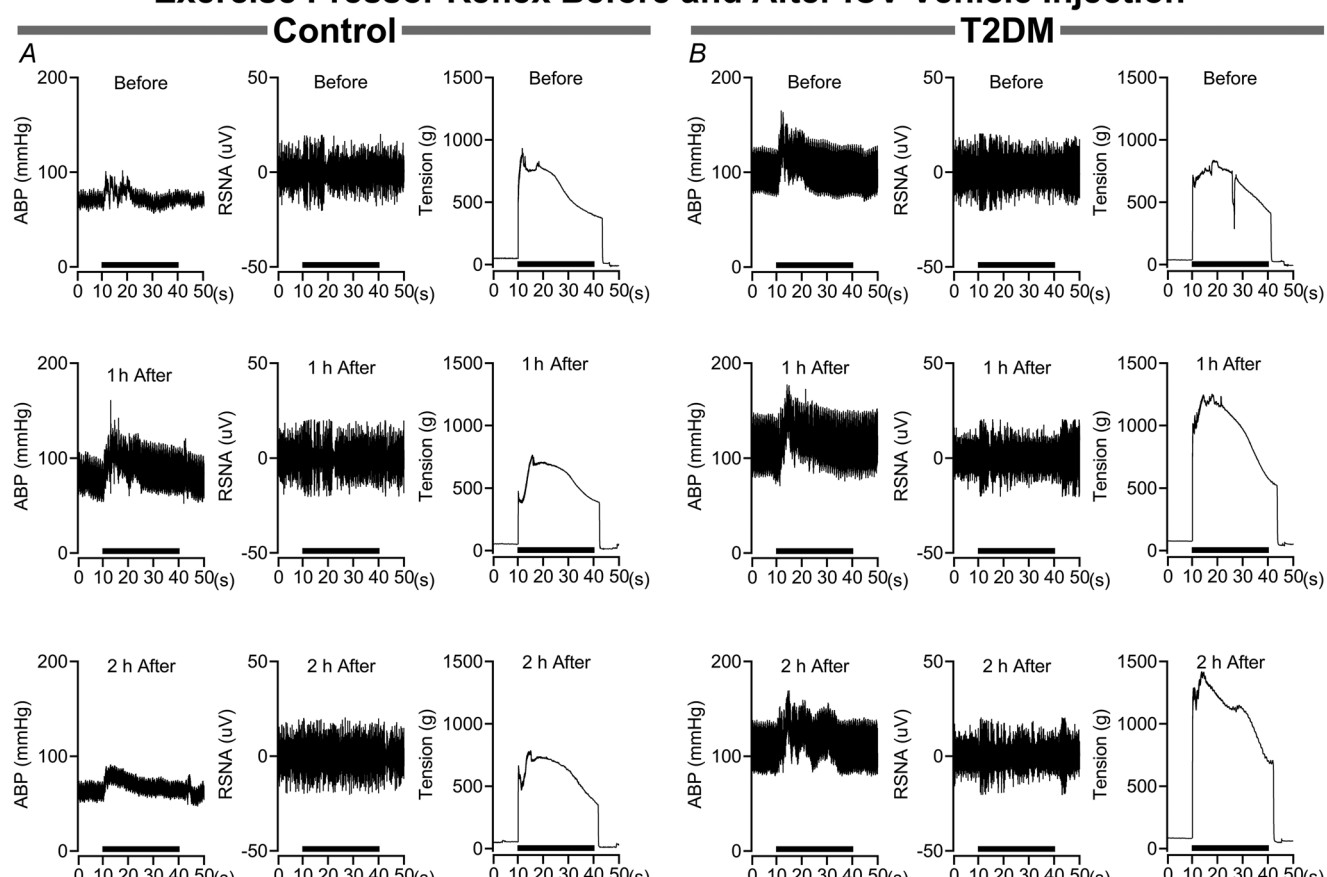

**Figure 3. Influence of intracerebroventricular vehicle injections on the exercise pressor reflex in control and T2DM rats**
*A*, in control rats brain vehicle injection has no influence on the arterial blood pressure (ABP) and RSNA responses during the EPR stimulation period. *B*, in T2DM rats the ABP and RSNA responses during the EPR stimulation are not influenced by vehicle injections. The EPR stimulation period (30 s) is represented by the black bar above the *x*-axis (seconds). Data are raw blood pressure, RSNA and muscle tension tracings before and after i.c.v. vehicle injections.

could not be resolved, and therefore only the area within 260–25 kDa on each membrane is shown. Due to limited samples the blot for insulin receptor $\beta$-subunit (Fig. 5*E*) was developed after stripping and re-probing a previous blot, and the dark bands below the protein of interest are the result of incompletely stripped antibodies and or interactions of fresh antibodies with those incompletely stripped antibodies.

In Fig. 6, western blot experiments demonstrated that insulin signalling is disrupted to some extent in the NTS of T2DM rats. Baseline expression of phospho-PI3K was significantly decreased in T2DM relative to control rats (1.00 ± 0.24 a.u. in control *vs.* 0.80 ± 0.07 a.u. in T2DM, $P = 0.032$ for unpaired *t* test), while the corresponding expression of total PI3K was not statistically different between groups (1.00 ± 0.19 a.u. in control *vs.* 1.13 ± 0.19 a.u. in T2DM). Furthermore, the ratio of phospho-PI3K to total PI3K was significantly decreased in T2DM relative to control rats (1.00 ± 0.26 a.u. in control *vs.* 0.77 ± 0.12 a.u. in T2DM, $P = 0.031$ for unpaired *t* test). Additionally, the baseline expression of phospho-Akt was not different between groups (1.00 ± 0.21 a.u. in control

*vs.* 1.03 ± 0.16 a.u. in T2DM), while the corresponding expression of total Akt was likewise not different between groups (1.00 ± 0.12 a.u. in control *vs.* 1.04 ± 0.23 a.u. in T2DM). Furthermore, the ratio of phospho-Akt to total Akt did not differ between groups (1.00 ± 0.26 a.u. in control *vs.* 0.91 ± 0.17 a.u. in T2DM). Expression of the IR-$\beta$-subunit was not different between groups (1.00 ± 0.28 a.u. in control *vs.* 0.94 ± 0.26 a.u. in T2DM).

## Discussion

The current investigation revealed several key findings in the low-dose STZ/HFD rat model of T2DM. (1) Direct injections of insulin into the brain significantly attenuated the augmented EPR response in T2DM relative to control male rats. (2) In fasting rats basal plasma insulin levels were decreased in T2DM as compared to control rats. (3) In fasting rats basal cerebrospinal fluid insulin levels were also lower in T2DM than in control rats. (4) Finally, in fasted rats basal PI3K signalling was disrupted in the NTS of T2DM in comparison with control rats, and in parallel with reductions in CSF insulin.

## ICV Vehicle Injections

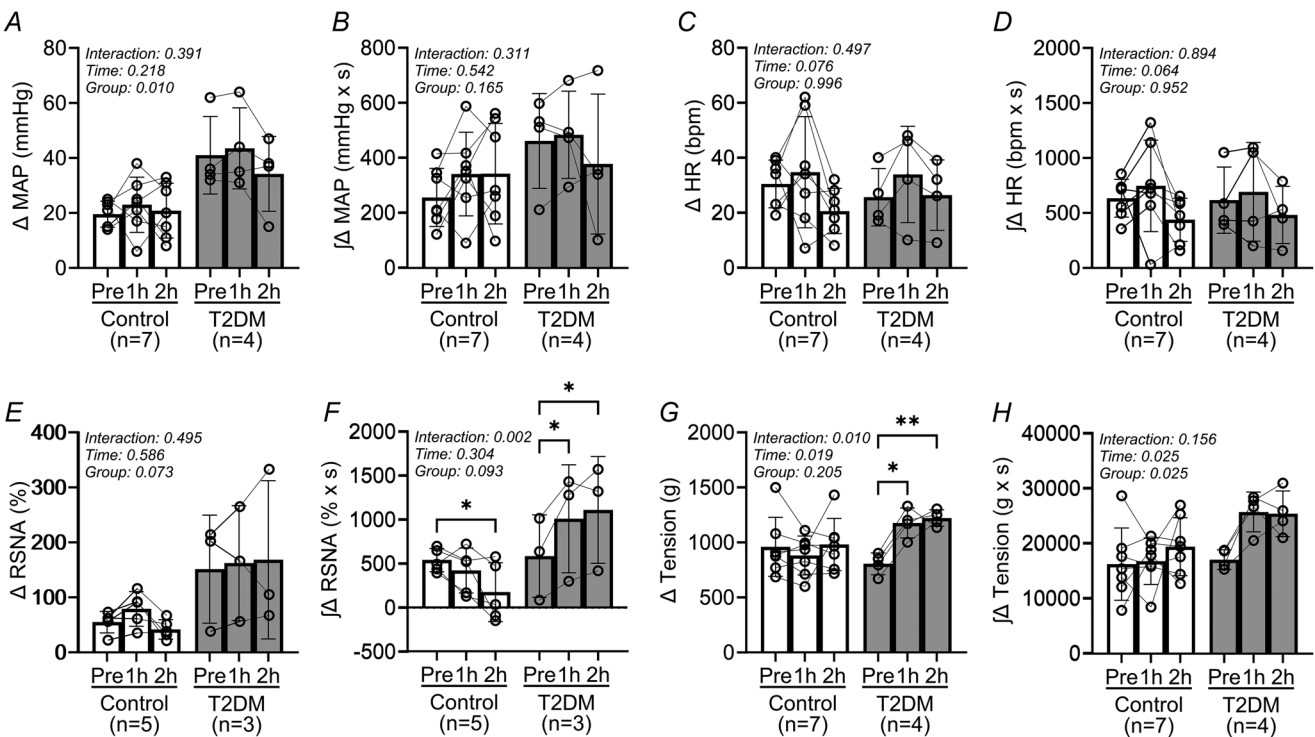

**Figure 4. Intracerebroventricular vehicle injections do not influence the exercise pressor reflex in control and T2DM rats**
Vertical bars represent the mean pressor (*A* and *B*), cardioaccelerator (*C* and *D*), and sympathetic (*E* and *F*) responses, and hindlimb muscle tension (*G* and *H*) during EPR stimulation. These variables were measured before (Pre) and after (1 and 2 h) I.C.V. vehicle injections in control (open bars and circles) and T2DM (filled bars and circles) rats. Analysed by two-way RM ANOVA and Šidák's multiple comparison test. \**P* < 0.05, \*\**P* < 0.01. Bars are the mean ± SD.

Our previous investigation demonstrated that the EPR is deleteriously augmented in the STZ/HFD rat model of T2DM (Kim et al., 2019). The current study produced almost identical results, with pressor and sympathetic responses to EPR activation augmented in the T2DM rats relative to the control group (Figs 2*A*, *B* and 4*A*, *B*). However, the cardioaccelerator responses to EPR activation did not appear augmented relative to the control group (Figs 2*C*, *D* and 4*C*, *D*). However, we have previously demonstrated that the low-dose STZ/HFD rat model may produce a phenotype with augmented pressor and sympathetic responses to hindlimb stimulation with I.A. capsaicin, but no augmented cardioaccelerator responses relative to control animals (Ishizawa et al., 2021). Additionally, although fasting blood glucose was elevated in T2DM rats, fasting plasma insulin decreased in the T2DM relative to control rats in the present study (Table 1). This is consistent with previous findings using the UC-Davis T2DM rat model, whereas it was shown that fasting plasma insulin may be decreased in early, established and chronic T2DM (Huo et al., 2022). Furthermore, the difference in body weight between control and T2DM rats was negligible (Table 1). Finally, in the T2DM rats blood glucose remained significantly elevated throughout the experimental period (Table 2). In STZ/HFD rats the elevations in fasting blood glucose in the conscious state and post-anaesthesia exposure, as well as the observed central hypoinsulinaemia, were the most delineating factors determining the presence of T2DM. Together, these findings demonstrate successful

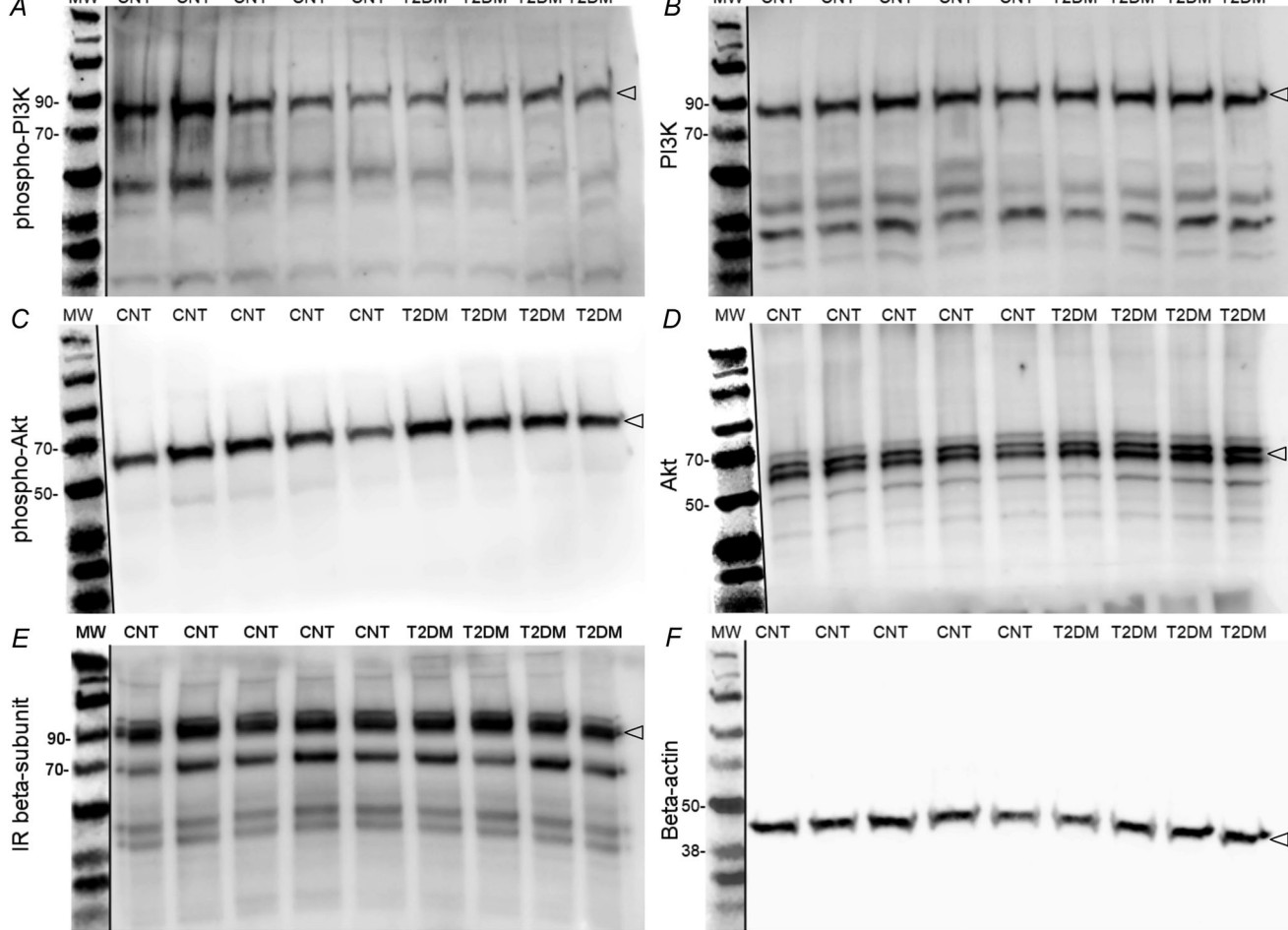

**Figure 5. Western blot images of NTS micropunches from T2DM and control rats**
Individual lanes represent NTS micropunch samples from a single animal. Each panel illustrates the area of the membrane probed by primary antibodies for target proteins. Target proteins on each membrane are phosphorylated phosphoinositide 3-kinase (phospho-PI3K; *A*), PI3K (*B*), phosphorylated Akt (phospho-Akt; *C*), Akt (*D*), insulin receptor β-subunit (IR β-subunit; *E*), and β-actin (*F*) as the loading control. Protein ladders between 260 and 25 kDa were resolved and select molecular mass markers are labelled. Open arrowheads (right edge on membranes) indicate the bands used for quantification. Bands were developed using enhanced chemiluminescence (*A–E*) or fluorescence (*F* and molecular mass markers). Fluorescence images were converted to greyscale images, and protein ladders were cropped into each membrane.

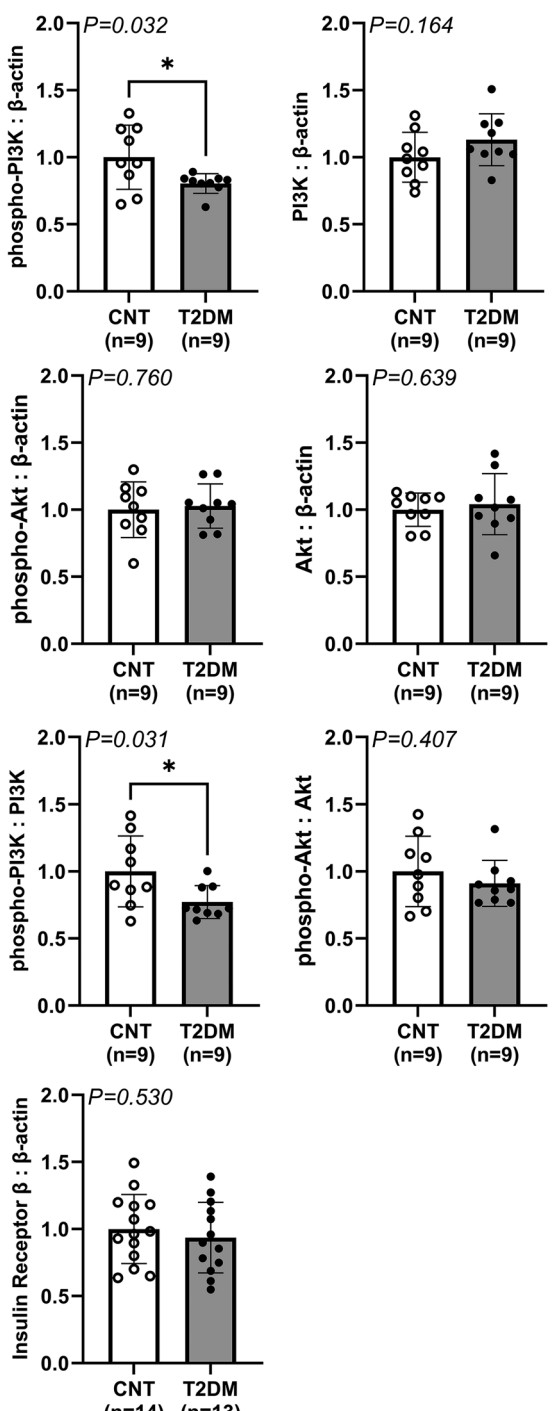

**Figure 6. Expression of insulin signalling related proteins in the NTS of fasted T2DM and control rats**

Vertical bars represent the mean normalized optical densities for target proteins in control (CNT; open bars and circles) *versus* T2DM (dark bars and circles) rats. Target proteins are phosphoinositide 3-kinase (PI3K), phospho-PI3K (p-PI3K), serine/threonine kinase Akt (Akt), phospho-Akt (p-Akt), insulin receptor $\beta$-subunit (IR) and $\beta$-actin. Analysed by unpaired *t* test. *$P < 0.05$. Bars are the mean $\pm$ SD.

generation of the T2DM phenotype in the low-dose STZ/HFD rats used in the present study.

In the present study, we performed microinjections of insulin or vehicle solutions into the brain to test the hypothesis that the augmented EPR in T2DM rats can be attenuated by partial restoration of brain insulin levels. In the I.C.V. insulin experiments we observed that the abnormal pressor and sympathetic responses in T2DM rats decreased analogously with each other over time following insulin injections (Fig. 2). The same trend was present in the cardioaccelerator response to EPR activation following insulin injections in the T2DM rats (Fig. 2*C, D*). In contrast, activation of the EPR did not influence the pressor, sympathetic, or cardioaccelerator responses in the I.C.V. vehicle experiments, in either the control or the T2DM groups (Fig. 4). Additionally, the resting blood pressure in T2DM rats decreased significantly following insulin microinjections, but not in control rats. Collectively, these findings demonstrate that microinjections of insulin into brain cerebrospinal fluids normalize the EPR in T2DM rats. This finding is consistent with previous work that demonstrates potentiation of the EPR following pharmacological blockade of endogenous insulin signalling in the NTS (Estrada et al., 2023). That is, blockade of endogenous insulin signalling in healthy rats may mirror the low CSF insulin in T2DM rats, and activation of brain insulin signalling in T2DM would restore the normal responses to EPR activation, which indeed is the case. These results are further corroborated by previous observations where it was demonstrated that I.C.V. insulin injections attenuate pressor responses evoked by stimulation from multiple brain areas, including the rostral ventrolateral medulla (Kuo et al., 1993), which receives sensory information from activation of the EPR (Potts et al., 2002).

The NTS is a hindbrain region that receives and integrates sensory information from muscle sensory fibres in the EPR pathway and is also a critical site for the integration of other cardiovascular peripheral inputs (Potts et al., 2002). For example, we recently demonstrated increased c-Fos activation of IR-positive neurons within the NTS following repetitive stimulation of the EPR (Estrada et al., 2023). Therefore, we examined NTS micropunches from both control and T2DM rats to determine if T2DM alters IR signalling within this region of the hindbrain. Protein expression of the IR and its canonical signalling enzymes (IR-$\beta$ subunit, total and phosphorylated PI3K, total and phosphorylated Akt) were evaluated by western blot densitometry (Fig. 5*A*). Here, our evaluation of PI3K in the NTS demonstrated modest but significant deficits in its phosphorylated expression and inferred basal activity (phospho-PI3K:PI3K) in T2DM (Fig. 5*B*). This finding is consistent with the reductions in CSF insulin found

in T2DM rats. Thus, the augmented EPR observed in T2DM rats is associated with reductions in brain insulin and concomitant disruptions in brain IR signalling, specifically phospho-PI3K. Future investigations should include whether IR signalling is disrupted in both sympathoexcitatory and sympathoinhibitory NTS neurons in T2DM. Additionally, further work is also necessary to confirm whether brain insulin signalling within other cardiovascular control neurons, such as the rostral ventrolateral medulla, is also affected in T2DM animals (Kuo et al., 1993).

The finding of decreased brain insulin and disrupted PI3K signalling in the NTS of T2DM rats is in line with the hypothesis that central hypoinsulinaemia results in altered excitability of neurons in the cardiovascular control areas of the hindbrain. It was previously demonstrated that insulin suppresses the firing of appetite-controlling hypothalamic proopiomelanocortin neurons, through IR-mediated phosphorylation of PI3K and activation of $K_{ATP}$ channels (Plum et al., 2006). Here, we observed disrupted PI3K phosphorylation in the NTS of T2DM rats thereby supporting a similar role for insulin in this context. That is, insulin may regulate the activity of the cardiovascular control neurons that modulate the EPR, via the IR/PI3K pathway. Further downstream of IR stimulation, the enzyme Akt is activated by PI3K. Here, we did not observe significant decreases in the phosphorylated expression of Akt or in its inferred baseline activity (phospho-Akt:Akt) (Fig. 5*B*). Furthermore, insulin signalling through the IR/PI3K/Akt pathway is instead implicated in the regulation of glucose and lipid metabolism in peripheral tissues (Huang et al., 2018). Diseases such as Alzheimer's disease produce changes in central insulin sensitivity, which can occur in part because of changes in IR expression (Sedzikowska & Szablewski, 2021). However, IR expression was not observed to have changed in NTS micropunches from T2DM rats (Fig. 5*B*), suggesting that the low-dose STZ/HFD rats used in this study may not have altered central insulin sensitivity. Thus, our evaluation of protein expression here remains congruent with a role for IR-signalling in regulating NTS neuronal excitability. Collectively, results from analysis of CSF insulin and NTS protein expression in control and T2DM fasted rats suggest a role for the IR/PI3K pathway in modulating the EPR.

The present finding that CSF insulin is decreased in the STZ/HFD rat model of T2DM merits further discussion. Previous studies demonstrate that insulin binds to brain endothelial cells and the density of insulin on these cells may be impaired in T2DM (Kaiyala et al., 2000; Schwartz et al., 1990). The precise mechanisms of insulin transport are still not fully resolved. However, the process requires active transport of insulin via insulin transport protein(s), which transports insulin into the CSF (Banks et al., 2012). This process is saturable and can be

regulated by insulin itself and other pathophysiological factors such as triglycerides and hyperglycaemia (Banks et al., 2012), as well as other serum factors (Brown et al., 2022). Thus, brain insulin has a close relationship with peripheral insulin and the diabetic state (Banks et al., 2012). Although the findings in the present investigation could not determine if peripheral hyperinsulinaemia is associated with central hypoinsulinaemia, the finding of reduced CSF insulin is still consistent with experimental expectations in T2DM rats (Huo et al., 2022).

Collectively, the current findings expand our understanding of how T2DM may influence cardiovascular function. However, further consideration should be given as multiple factors influence the overall phenotype of the STZ/HFD rat including age of rats, dietary composition and duration following STZ injections, STZ dose and number of STZ injections (Skovso et al., 2014). These factors together determine the residual $\beta$-cell mass and degree of aberrations in insulin signalling the animal presents with. The STZ/HFD rat phenotype can thus reflect early- through late-stage T2DM (Skovso et al., 2014). Investigating the precise differences in T2DM stage is an extensive undertaking, and beyond the scope of this study. Nonetheless, the T2DM phenotype in the present study may resemble the established-stage of T2DM (Huo et al., 2022). The implications, here, are that central insulin treatment may be used to normalize abnormal EPR responses to exercise in T2DM up to at least the early established stage of the disease.

The present study is limited in some respects. First, as the current study builds upon our previous work done in male rats only (Ishizawa et al., 2021; Kim et al., 2019), we excluded the use of female rats in the present study because we have not yet established whether our low-dose STZ/HFD protocol used to generate the T2DM phenotype in male Sprague–Dawley rats also produces an exaggerated EPR in female rats. Next, we have acknowledged that fasting plasma insulin levels are decreased in the T2DM relative to the control group. Thus, the rats used in the current study may reflect an established stage of T2DM (Huo et al., 2022; Skovso et al., 2014). We only measured fasting insulin levels on the day of terminal experiments, and hyperinsulinaemia may have been present at an earlier time point during the study as we previously demonstrated (Ishizawa et al., 2021; Kim et al., 2019). However, this does not detract from the current investigation as the main hypothesis was tested. Future investigations should track fasting glucose and insulin periodically throughout the feeding window to determine when terminal experiments should be conducted. This would further ensure validation of the STZ/HFD phenotype, i.e. early/established/late-stage T2DM, and more extensive hypothesis testing. A third limitation was the number of samples used in the comparison of CSF insulin between control and T2DM

rats. Only four samples for the T2DM group were within detectable range of the assay. However, we contend that this outcome further supports our hypothesis that low CSF insulin is characteristic in this model of T2DM. Fourth, we observed inconsistencies in the sympathetic responses to EPR activation in the i.c.v. vehicle experiments (Fig. 4) compared with the i.c.v. insulin experiments (Fig. 2) in both the control and T2DM rats. However, both the number of animals and successful RSNA recordings were limited in the i.c.v. vehicle data analysis. Inspection of the data in both i.c.v. insulin (Fig. 2) and i.c.v. vehicle (Fig. 4) experiments further reveals that both the peak and integrated pressor responses to EPR activation before microinjections were consistent with previous findings (Kim et al., 2019). Furthermore, the pressor and sympathetic responses to EPR activation following i.c.v. microinjections of insulin and aCSF followed the expected experimental outcomes. Lastly, we did not measure brain insulin levels following i.c.v. injections of insulin, nor did we measure changes in phosphorylated PI3K or Akt after the injections as this was beyond the scope of the current study. However, based on previous work the precise half-life of insulin in the brain has not been determined (Grey & Barrett, 2018). We performed bolus injections of exogenous insulin (500 mU) into the brain based on the significant increases seen in brain insulin and long-lasting physiological actions over 3 h without affecting plasma insulin, and brain or blood glucose levels (Rhamouni et al., 2004).

In summary, the major findings of the current study suggest that T2DM results in aberrations in blood glucose and insulin, and insulin signalling within the central nervous system that coincide with augmented EPR function. That is, the amount of insulin available in the CNS may directly impact neuronal excitability within the cardiovascular control centres, such as the NTS, thereby influencing cardiovascular function (Mizuno et al., 2021). In support of this, peripheral hyperglycaemia and central hypoinsulinaemia were associated with an augmented EPR in T2DM rats. Furthermore, although the T2DM rats in this study had peripheral hypoinsulinaemia, the EPR may be augmented without concurrent elevations in both peripheral blood glucose and plasma insulin, as previously demonstrated (Huo et al., 2022). Treatment with direct injections of insulin into the brain normalized the augmented EPR responses in T2DM rats. Additionally, the expression of IR-positive neurons in the NTS is ubiquitous, and insulin signalling within these neurons may maintain the capacity to buffer the EPR (Estrada et al., 2023). Thus, it is conceivable that disrupted PI3K signalling in the NTS may be one of several factors underlying the exaggerated EPR in T2DM rats.

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

## Additional information

### Data availability statement

The data supporting the present findings are available from the corresponding author upon reasonable request. Due to privacy or ethical restrictions, the data is not publicly available.

## Competing interests

The authors declare no conflict of interest.

## Author contributions

J.A.E. and M.M. conceived and designed experiments. J.A.E., R.I. and M.M. performed experiments. J.A.E. analysed the data. J.A.E., R.I., H.-K.K., A.F., A.H., N.H., G.A.I., S.A.S., W.V. and M.M. drafted the manuscript. All authors were involved in revising the manuscript. All authors have read and approved the final version of this manuscript and agree to be accountable for all aspects of the work in ensuring that questions related to the accuracy or integrity of any part of the work are appropriately investigated and resolved. All persons designated as authors qualify for authorship, and all those who qualify for authorship are listed.

## Funding

This work was supported by the National Heart, Lung, and Blood Institute (R01HL-151632) (to M.M.), National Heart, Lung, and Blood Institute Diversity Supplement Award 3R01HL151632-01S1 (to J.A.E.); and JSPS KAKENHI Grant Number JP 20H04083 (to N.H.)

## Acknowledgements

We would like to acknowledge the late Dr Jere H. Mitchell for his contributions and support in successfully getting this project off the ground.

## Keywords

blood pressure, exercise pressor reflex, insulin, nucleus tractus solitarius, PI3K, sympathetic activity

## Supporting information

Additional supporting information can be found online in the Supporting Information section at the end of the HTML view of the article. Supporting information files available:

**Peer Review History**

