## [Peer Review History · The Journal of Physiology]

Intracerebroventricular insulin injection acutely normalizes the augmented exercise pressor reflex in male rats with type 2 diabetes mellitus

Juan A Estrada, Rie Ishizawa, Han Kyul Kim, Ayumi Fukazawa, Amane Hori, Norio Hotta, Gary A Iwamoto, Scott A. Smith, Wanpen Vongpatanasin, and Masaki Mizuno

DOI: 10.1113/JP286715

Corresponding author(s): Masaki Mizuno (masaki.mizuno@utsouthwestern.edu)

Review Timeline:

Submission Date:	09-Apr-2024
Editorial Decision:	26-Apr-2024
Revision Received:	23-May-2024
Editorial Decision:	14-Jun-2024
Revision Received:	02-Jul-2024
Accepted:	15-Jul-2024

Senior Editor: Vaughan Macefield

Reviewing Editor: Marc Kaufman

Transaction Report:

Dear Dr Mizuno,

Re: JP-RP-2024-286715 "Intracerebroventricular insulin injection acutely normalizes the augmented exercise pressor reflex in a rat model of type 2 diabetes mellitus" by Juan A Estrada, Rie Ishizawa, Han Kyul Kim, Ayumi Fukazawa, Amane Hori, Norio Hotta, Gary A Iwamoto, Scott A. Smith, Wanpen Vongpatanasin, and Masaki Mizuno

Thank you for submitting your manuscript to The Journal of Physiology. It has been assessed by a Reviewing Editor and by 2 expert referees and we are pleased to tell you that it is acceptable for publication following satisfactory revision.

REVISION CHECKLIST:

Please upload two versions of your manuscript text: one with all relevant changes highlighted and one clean version with no changes tracked. The manuscript file should include all tables and figure legends, but each figure/graph should be uploaded as separate, high-resolution files. The journal is now integrated with Wiley's Image Checking service. For further details, see: <https://www.wiley.com/en-us/network/publishing/research-publishing/trending-stories/upholding-image-integrity-wileys-image-screening-service>.

- 'Potential Cover Art' for consideration as the issue's cover image
- Appropriate Supporting Information (video, audio or data set: see https://jp.msubmit.net/cgi-bin/main.plex?form_type=display_requirements#supp)

We look forward to receiving your revised submission.

Yours sincerely,

Vaughan Macefield
Senior Editor
The Journal of Physiology

REQUIRED ITEMS

- Author photo and profile. First or joint first authors are asked to provide a short biography (no more than 100 words for one author or 150 words in total for joint first authors) and a portrait photograph. These should be uploaded and clearly labelled together in a Word document with the revised version of the manuscript. See Information for Authors for further details.

- The reference list must be in alphabetical order, rather than numbered, to comply with our Journal format.

- Your manuscript must include a complete Additional Information section, including competing interests; funding; author contributions and acknowledgements.

- You must upload original, uncropped western blot/gel images (including controls) if they are not included in the manuscript. This is to confirm that no inappropriate, unethical or misleading image manipulation has occurred. These should be uploaded as 'Supporting information for review process only'. Please label/highlight the original gels so that we can clearly see which sections/lanes have been used in the manuscript figures. For more information, see: <https://physoc.onlinelibrary.wiley.com/hub/journal-policies#imagmanip>.

- Papers must comply with the Statistics Policy: https://jp.msubmit.net/cgi-bin/main.plex?form_type=display_requirements#statistics.

In summary:

- If $n \leq 30$, all data points must be plotted in the figure in a way that reveals their range and distribution. A bar graph with data points overlaid, a box and whisker plot or a violin plot (preferably with data points included) are acceptable formats.

- If $n > 30$, then the entire raw dataset must be made available either as supporting information, or hosted on a not-for-profit repository, e.g. FigShare, with access details provided in the manuscript.

- 'n' clearly defined (e.g. x cells from y slices in z animals) in the Methods. Authors should be mindful of pseudoreplication.

- All relevant 'n' values must be clearly stated in the main text, figures and tables.

- The most appropriate summary statistic (e.g. mean or median and standard deviation) must be used. Standard Error of the Mean (SEM) alone is not permitted.

- Exact p values must be stated. Authors must not use 'greater than' or 'less than'. Exact p values must be stated to three significant figures even when 'no statistical significance' is claimed.

- Please include an Abstract Figure file, as well as the Figure Legend text within the main article file. The Abstract Figure is a piece of artwork designed to give readers an immediate understanding of the research and should summarise the main conclusions. If possible, the image should be easily 'readable' from left to right or top to bottom. It should show the physiological relevance of the manuscript so readers can assess the importance and content of its findings. Abstract Figures should not merely recapitulate other figures in the manuscript. Please try to keep the diagram as simple as possible and without superfluous information that may distract from the main conclusion(s). Abstract Figures must be provided by authors no later than the revised manuscript stage and should be uploaded as a separate file during online submission labelled as File Type 'Abstract Figure'. Please also ensure that you include the figure legend in the main article file. All Abstract Figures should be created using BioRender. Authors should use The Journal's premium BioRender account to export high-resolution images. Details on how to use and access the premium account are included as part of this email.

EDITOR COMMENTS

Reviewing Editor:

Methods Details:

The authors should state how the STZ was injected into the rats. I assume that it was IP; is that correct?

Ethics Concerns:

No concerns

Comments to the Author:

The manuscript by Estrada et al has shown that in decerebrated rats that insulin injected into the fourth ventricle of the brain restored to normal levels the exaggerated exercise pressor reflex seen in rats made diabetic with streptozocin injections. Both reviewers, who have a large amount of experience with this preparation, found the manuscript well written and experiments described in it to be well controlled. In addition, both reviewers believed that the findings reported in the manuscript are novel and will significantly advance the field of control of the circulation during exercise. My only concern, and it is minor is that the authors need to clearly describe the source of their rats. Maybe I missed it but I could not find this information in the manuscript.

A major concern involves the western blots shown in figure 5. The entire ladder for each protein measured should be shown. I realize that this might require the addition of another figure or two. I realize that this will require the addition of another figure or two, but doing so will increase credibility. Last, I urge you to consider strongly both reviewers' requests to relate the concentration of insulin created in the fourth ventricle of the brain with physiological or pathophysiological levels.

Senior Editor:

Comments to the Author:

Thank you for submitting your manuscript to the Special Issue of The Journal of Physiology. Your manuscript has been assessed by two independent reviewers and a handling editor, all of whom are experts in the field. While all see merit in your study, and believe it will contribute to our understanding of the neural control of sympathetic outflow during exercise, as you will see there are several areas of concern that I will need you to address. Please provide point-by-point responses to each of the comments. I look forward to receiving your revised manuscript in due course.

Yours sincerely,

Vaughan Macefield

REFEREE COMMENTS

Referee #1:

This mechanistic study by Estrada et al sought to determine the effects of intracerebroventricular insulin injection on the exaggerated exercise pressor reflex in type 2 diabetic rats. The study was well conducted, and the manuscript is written clearly. I have a few minor comments below.

1. Line 111: Huo et al. did not measure afferent sensitivity. This statement should be reworded.
2. Line 125: "may be" should be "is."

3. Line 143: please explain why only male rats were used in this study. Also, the title should reflect that these findings were in male rats and do not apply in general to all with type 2 diabetes.
4. Line 147: what was the diagnostic criteria for type 2 diabetes? How did you determine they were diabetic?
5. Line 187: you have an extra ")"
6. Line 369: and individual data points
7. Line 419: the word "treatment" is confusing here as it sounds like the rats were treated for their insulin concentrations. Injection would be easier to understand or reword the sentence.
8. First paragraph of discussion could use more details. For example, were they resting plasma insulin levels or those during or after exercise?
9. Line 571: This connection is vital to your study and I think it would be very helpful if a schematic was provided at some point in the manuscript to clearly show the relationship you are suggesting.
10. Line 241: How was the dose of ICV insulin determined? Is this similar to what would be produced endogenously in healthy rats?

Referee #2:

Estrada and colleagues demonstrated that a bolus ICV administration of insulin fully normalized augmented RSNA and pressor responses to hindlimb skeletal muscle contraction in the rat T2DM model. Additionally, they found a decrease in the expression of phosphorylated PI3K was decreased in the NTS of T2DM compared to healthy controls. These observations led the authors to conclude that brain injection with insulin is capable of correcting abnormal EPR in T2DM and to further suggest a regulatory role of brain insulin in the EPR. It is interesting that a short-time increase in insulin in the brain can treat the abnormal EPR for a longer period. The experiments were expertly performed, and data have been presented in a clear and comprehensive manner in the manuscript. While I acknowledge the significant scientific contributions of this study in advancing our understanding of neural control of the circulation in health and disease, I would like to offer several comments for the authors' consideration.

1. I kindly request the authors to describe the rationale behind determining the amount of insulin (500 mU, 50 nL) administered ICV. How are the concentrations of CSF insulin estimated immediately, 1 hour, and 2 hours after administration? Do you assume that these concentrations are within the physiological range? If so, what evidence supports this assumption?
2. Related to my previous comment, another concern of mine is the short half-life of insulin. Are there any data or knowledge available to understand mechanisms underlying the effect of bolus, but not continuous, ICV administration of insulin to normalize the EPR in T2DM? Could you please address how acute insulin administration reduced the EPR responses in T2DM one or two hours after administration when the exogenous insulin had already almost disappeared? These questions arise since the causal relationships among the insulin concentration in the brain, PI3K pathways, and the EPR over experimental time have not been established.
3. The authors may hypothesize that the decreased expression of phosphorylated PI3K in T2DM in the NTS, an afferent baroreflex pathway, would underlie augmented EPR, likely through a decrease in the effect of PI3K pathway to hyperpolarize the NTS neurons via KATP channels activation (Plum et al. 2006). However, the NTS includes excitatory and inhibitory, i.e., sympathoinhibitory and sympathoexcitatory, neurons. Glutamate injection into the NTS decreases arterial pressure. Are there any clues suggesting that the expression of phosphorylated PI3K is decreased in sympathoexcitatory NTS neurons in T2DM or in both neuronal populations of T2DM?

Other comments are as follows:

4. While the NTS is important for cardiovascular regulation, other critical regions or nuclei in the brainstem should be noted. Why did the authors focus only on the NTS?
5. Figure 1 and 3: Please include the muscle tension tracings because reflex responses cannot be adequately compared without considering muscle tension.

6. Figure 2 and 4: Individual data have been presented as dot plots. However, for a clearer demonstration of the observation, please consider adding connecting lines between the plots within an individual animal among Pre, 1h, and 2h.
7. Figure 2: Individual plots indicate that peak tension generated by muscle contraction was around 200 g in some rats after ICV administration. The decrease in tension over experimental time may suggest deteriorations of the preparation. Do authors consider the data obtained from these preparations reasonable for data presentation?

END OF COMMENTS

Confidential Review

09-Apr-2024

Intracerebroventricular insulin injection acutely normalizes the augmented exercise pressor reflex in male rats with type 2 diabetes mellitus

Senior Editor:

Thank you for submitting your manuscript to the Special Issue of The Journal of Physiology. Your manuscript has been assessed by two independent reviewers and a handling editor, all of whom are experts in the field. While all see merit in your study, and believe it will contribute to our understanding of the neural control of sympathetic outflow during exercise, as you will see there are several areas of concern that I will need you to address. Please provide point-by-point responses to each of the comments. I look forward to receiving your revised manuscript in due course.

Author Response: We would like to thank each of the editors and referees for the additional comments and suggestions. We believe that the resubmitted manuscript is much improved following incorporation of the suggested revisions. We have addressed all comments as completely as possible in the following responses to reviewers. All changes made to the manuscript are indicated in red font in the red-lined version of the manuscript. In addition, the specific pages and lines in which changes have been made are listed in the responses.

Reviewing Editor:

Methods Details: The authors should state how the STZ was injected into the rats. I assume that it was IP; is that correct?

Author Response: The reviewing editor is correct in assuming the route of STZ administration was via an intraperitoneal injection. We have made a few minor changes within the text so that the route of STZ administration is clear for readers (Lines 146 and 149-150).

The manuscript by Estrada et al has shown that in decerebrated rats that insulin injected into the fourth ventricle of the brain restored to normal levels the exaggerated exercise pressor reflex seen in rats made diabetic with streptozotocin injections. Both reviewers, who have a large amount of experience with this preparation, found the manuscript well written and experiments described in it to be well controlled. In addition, both reviewers believed that the findings reported in the manuscript are novel and will significantly advance the field of control of the circulation during exercise. My only concern, and it is minor is that the authors need to clearly describe the source of their rats. Maybe I missed it but I could not find this information in the manuscript.

Author Response: We have introduced the source of the rats in the first sentence as suggested (Lines 144-145). Thank you.

A major concern involves the western blots shown in figure 5. The entire ladder for each protein measured should be shown. I realize that this might require the addition of another figure or two. I realize that this will require the addition of another figure or two, but doing so will increase credibility.

Author Response: We completely agree with the reviewing editor that adding western blot images with molecular weight ladders included will add credibility to the results. As suggested, Figure 5 has been updated to illustrate the area corresponding to all resolved protein ladders (260 – 25 kDa, or 9 of 12 molecular weight ladders). Proteins below 25kDa could not be resolved with the 10% pre-cast gels we used, and therefore this region of the membrane (the lowest portion) was cropped out of each image. Each image in the figure faithfully represents the results for each primary antibody used. Furthermore, our protein ladder was labelled with a fluorescent dye therefore we converted each ladder into a grey scale image and then overlaid the ladder into the images. Additionally, we have added another figure (Fig. 6) to show the results of quantification. The original uncropped images have been uploaded as 'Supporting information for review process only'. In accordance with the above, we have updated the manuscript (Lines 310-312, 335-336, 374-381, 508-513, 908-925).

Last, I urge you to consider strongly both reviewers' requests to relate the concentration of insulin created in the fourth ventricle of the brain with physiological or pathophysiological levels.

Author Response: We have provided a commentary regarding brain insulin levels following ICV injections of insulin as a study limitation (Lines 670-677).

Referee #1:

This mechanistic study by Estrada et al sought to determine the effects of intracerebroventricular insulin injection on the exaggerated exercise pressor reflex in type 2 diabetic rats. The study was well conducted, and the manuscript is written clearly. I have a few minor comments below.

1. Line 111: Huo et al. did not measure afferent sensitivity. This statement should be reworded.

Author Response: Thank you for highlighting this mistake. We have removed the citation (Line 111).

2. Line 125: "may be" should be "is."

Author Response: Thank you for the correction, the text has been edited (Line 126).

3. Line 143: please explain why only male rats were used in this study. Also, the title should reflect that these findings were in male rats and do not apply in general to all with type 2 diabetes.

Author Response: We agree with the reviewer that the title should specify that these studies were only done in male rats. As suggested, we have revised the title as follows: "Intracerebroventricular insulin injection acutely normalizes the augmented exercise pressor reflex in male rats with type 2 diabetes mellitus" (Lines 1-2). In the present study, we only used male rats because we have not yet established whether our low-dose STZ/HFD protocol used to generate the T2DM phenotype in male Sprague-Dawley rats also produces an exaggerated exercise pressor reflex in female rats. The current study only builds upon our previous work done in male rats (Kim et al 2019; doi: 10.1152/ajpregu.00061.2019.). We have added description on this point in the revised manuscript (Lines 645-649). Thank you for bringing this to our attention.

4. Line 147: what was the diagnostic criteria for type 2 diabetes? How did you determine they were diabetic?

Author Response: Thank you for suggesting we clarify how we determine the successful generation of T2DM rats. We made several observations to determine whether our rats were diabetic. Reiterated shortly here, we assessed the averages for 1) fasting blood glucose to determine if STZ/HFD rats were hyperglycemic relative to controls (101.4 ± 13.6 mg/dL vs 178.3 ± 107.2 mg/dL, control vs T2DM), 2) fasting plasma insulin to determine if insulin levels change in HFD/STZ rats relative to controls (1.53 ± 0.40 ng/mL vs 0.91 ± 0.27 , control vs T2DM), 3) fasting CSF insulin to determine if the HFD/STZ rats are hypoinsulinemic relative to controls (0.41 ± 0.19 mg/dL vs 0.11 ± 0.05 mg/dL, control vs T2DM), 4) experimental blood glucose levels to determine if isoflurane exposure worsened hyperglycemia in STZ/HFD rats relative to controls (151.9 mg/dL vs 292.7 mg/dL, control vs T2DM) as has been shown in STZ/HFD mice models (Fang et al 2020; doi: 10.1016/j.brainres.2019.146511). We found that in each case, the averages were different between HFD/STZ and control rats. Furthermore, we found no significant differences in the average body weight of STZ/HFD rats relative to controls (499 ± 37 g vs 484 ± 45 g, control vs T2DM). We believe the most delineating factors were the blood glucose and CSF insulin comparisons. We have added this information in the methods section (Lines 263-264, 272-279, 401-402, 553-555).

5. Line 187: you have an extra ")"

Author Response: The text has been adjusted here for the readers (Line 188). Thank you.

6. Line 369: and individual data points

Author Response: We have added the suggestion to the text (Line 390). Thank you.

7. Line 419: the word "treatment" is confusing here as it sounds like the rats were treated for their insulin concentrations. Injection would be easier to understand or reword the sentence.

Author Response: Once again, we thank the reviewer for the correction here. The suggested change has been made (line 441).

8. First paragraph of discussion could use more details. For example, were they resting plasma insulin levels or those during or after exercise?

Author Response: We agree with the reviewer that more detail should be included in the first paragraph of the discussion to add context and specificity. We have indicated that the measured PI3K levels in control and T2DM fasted rats is the basal or non-insulin stimulated expression (Lines 533-536).

9. Line 571: This connection is vital to your study and I think it would be very helpful if a schematic was provided at some point in the manuscript to clearly show the relationship you are suggesting.

Author Response: Thank you for the suggestion. We have added a schematic as a part of the graphic abstract to put the western blot data and ELISA data into perspective, and updated the text accordingly (lines 616-618).

10. Line 241: How was the dose of ICV insulin determined? Is this similar to what would be produced endogenously in healthy rats?

Author Response: We used a previous study by Rahmouni et al 2004 (doi: [10.1172/JCI21737](https://doi.org/10.1172/JCI21737)) to determine the exogenous dose of insulin injected directly into the brain. We have cited the paper (Line 243-245).

Referee #2:

Estrada and colleagues demonstrated that a bolus ICV administration of insulin fully normalized augmented RSNA and pressor responses to hindlimb skeletal muscle contraction in the rat T2DM model. Additionally, they found a decrease in the expression of phosphorylated PI3K was decreased in the NTS of T2DM compared to healthy controls. These observations led the authors to conclude that brain injection with insulin is capable of correcting abnormal EPR in T2DM and to further suggest a regulatory role of brain insulin in the EPR. It is interesting that a short-time increase in insulin in the brain can treat the abnormal EPR for a longer period. The experiments were expertly performed, and data have been presented in a clear and comprehensive manner in the manuscript. While I acknowledge the significant scientific contributions of this study in advancing our understanding of neural control of the circulation in health and disease, I would like to offer several comments for the authors' consideration.

1. I kindly request the authors to describe the rationale behind determining the amount of insulin (500 mU, 50 nL) administered ICV. How are the concentrations of CSF insulin estimated immediately, 1 hour, and 2 hours after administration? Do you assume that these concentrations are within the physiological range? If so, what evidence supports this assumption?

Author Response: Based on a previous report, 500mU of exogenous insulin had been successfully used in acute ICV experiments several hours in duration (Rahmouni et al 2004, doi: [10.1172/JCI21737](https://doi.org/10.1172/JCI21737)) (Line 243-245). We used this dose based on this previous observation. However, it was beyond the scope of the present study to determine the concentrations of insulin over the experimental period, i.e. immediately and following ICV injections of insulin or vehicle solutions. This would have required us to generate more animals to sample CSF fluids from the cisterna magna at each time point for every group. Furthermore, while CSF continues to be a proxy for insulin concentrations within brain tissue the actual concentration of insulin within brain interstitial fluids may differ (Gray and Barrett 2018, doi: [10.1152/ajpcell.00240.2017](https://doi.org/10.1152/ajpcell.00240.2017)). We have added the relevant text in the discussion (Lines 670-677).

2. Related to my previous comment, another concern of mine is the short half-life of insulin. Are there any data or knowledge available to understand mechanisms underlying the effect of bolus, but not continuous, ICV administration of insulin to normalize the EPR in T2DM? Could you please address how acute insulin administration reduced the EPR responses in T2DM one or two hours after

administration when the exogenous insulin had already almost disappeared? These questions arise since the causal relationships among the insulin concentration in the brain, PI3K pathways, and the EPR over experimental time have not been established.

Author Response: The precise half-life of insulin in the brain is still unknown and insulin actions in the brain can remain elevated after ICV injections for several hours (Gray and Barrett 2018). We chose to perform a bolus dose of insulin as Rahmouni et al did, rather than a continuous infusion because that study suggests that RSNA activity would not be influenced significantly by insulin over the three-hour time course of the present study. Additionally, the extended period over which insulin may exert its actions may also be related to receptor isoform sensitivity. In adult mammals, isoform A of the insulin receptor is predominantly expressed in the brain and this isoform has two-fold higher affinity for insulin than isoform B which predominates in the periphery (Gray and Barrett 2018). Thus, brain tissue is more sensitive to insulin and even low doses of insulin (0.03mU) have been used for study in rats (Gray and Barrett 2018). The relevant text has been added to the text (Lines 670-677).

3. The authors may hypothesize that the decreased expression of phosphorylated PI3K in T2DM in the NTS, an afferent baroreflex pathway, would underlie augmented EPR, likely through a decrease in the effect of PI3K pathway to hyperpolarize the NTS neurons via KATP channels activation (Plum et al. 2006). However, the NTS includes excitatory and inhibitory, i.e., sympathoinhibitory and sympathoexcitatory, neurons. Glutamate injection into the NTS decreases arterial pressure. Are there any clues suggesting that the expression of phosphorylated PI3K is decreased in sympathoexcitatory NTS neurons in T2DM or in both neuronal populations of T2DM?

Author Response: The reviewer raises an interesting question. To the best of our knowledge, we do not know of any studies that demonstrate changes in PI3K activity in sympathoinhibitory/sympathoexcitatory NTS neurons in T2DM. We will add that this question needs further exploration in the manuscript (Lines 593-595).

Other comments are as follows:

4. While the NTS is important for cardiovascular regulation, other critical regions or nuclei in the brainstem should be noted. Why did the authors focus only on the NTS?

Author Response: We agree with the reviewer that the influence of other critical pathways in the brainstem should be considered. For the current study, we focused on the nuclei in the NTS mainly because our recent publication (Estrada et al 2023, doi.org/10.1096/fj.202300879RR) suggests that insulin signaling in this region plays a pivotal role in modulating the exercise pressor reflex. The results presented here are complementary to the previous report. Additionally, from a technical perspective micro-punches from the NTS are relatively easy to sample, as the central canal is easily visible and the contrast between gray matter (NTS) and white matter is also easily visualized. Furthermore, sampling other brainstem nuclei such as the rostral ventrolateral medulla would yield much less sample per rat thus requiring pooling of samples from multiple rats to perform successful western blots. This would require generating more T2DM and control rats to measure all the target insulin signaling related proteins. We have clarified why we focused on protein expression within the NTS in the last paragraph of the introduction (Lines 131-133).

5. Figure 1 and 3: Please include the muscle tension tracings because reflex responses cannot be adequately compared without considering muscle tension.

Author Response: We thank you for this good suggestion. The muscle tension tracings corresponding to the arterial blood pressure and renal sympathetic nerve activity tracings have been added for the readers (Fig. 1 and 3).

6. Figure 2 and 4: Individual data have been presented as dot plots. However, for a clearer demonstration of the observation, please consider adding connecting lines between the plots within an individual animal among Pre, 1h, and 2h.

Author Response: As suggested, we have added connecting lines for data points between individual animals. We thank the reviewer for the great suggestion to enhance our presentation of the data (Fig. 2 and 4).

7. Figure 2: Individual plots indicate that peak tension generated by muscle contraction was around 200 g in some rats after ICV administration. The decrease in tension over experimental time may suggest deteriorations of the preparation. Do authors consider the data obtained from these preparations reasonable for data presentation?

Author Response: We provide resting baseline blood pressures and blood glucose concentrations throughout the experimental period (Table 2) to assess the viability of the preparation. The tension decrease over time in some of the experiments was likely due to a decline in the excitability of the ventral root rather than deterioration of the animal. For example, previous works demonstrates that the overall quality and responsiveness of very similar preparations in decerebrate rats is stable over a 2 h experimental period. (Estrada and Kaufman 2018, doi.org/10.1152%2Fajpregu.00380.2017; Estrada et al 2023, doi.org/10.1096/fj.202300879RR). Furthermore, in Figures 2G and H, no significant main effect of time was observed. While there are indeed instances of decreased tension, they are not biased towards either group. In fact, no significant group-by-time were detected in the two-way ANOVA. Therefore, we included all the data for presentation after considering these factors.

Dear Dr Mizuno,

Re: JP-RP-2024-286715R1 "Intracerebroventricular insulin injection acutely normalizes the augmented exercise pressor reflex in male rats with type 2 diabetes mellitus" by Juan A Estrada, Rie Ishizawa, Han Kyul Kim, Ayumi Fukazawa, Amane Hori, Norio Hotta, Gary A Iwamoto, Scott A. Smith, Wanpen Vongpatanasin, and Masaki Mizuno

Thank you for submitting your manuscript to The Journal of Physiology. It has been assessed by a Reviewing Editor and by 2 expert referees and we are pleased to tell you that it is acceptable for publication following satisfactory revision.

REVISION CHECKLIST:

- 'Potential Cover Art' for consideration as the issue's cover image

- Appropriate Supporting Information (Video, audio or data set: see https://jp.msubmit.net/cgi-bin/main.plex?form_type=display_requirements#supp).

We look forward to receiving your revised submission.

Yours sincerely,

Vaughan Macefield
Senior Editor
The Journal of Physiology

REQUIRED ITEMS FOR REVISION

- You must upload original, uncropped western blot/gel images (including controls) if they are not included in the manuscript. This is to confirm that no inappropriate, unethical or misleading image manipulation has occurred. These should be uploaded as 'Supporting information for review process only'. Please label/highlight the original gels so that we can clearly see which sections/lanes have been used in the manuscript figures. For more information, see: <https://physoc.onlinelibrary.wiley.com/hub/journal-policies#imagmanip>.

- You must upload original, uncropped western blot/gel images (including controls) if they are not included in the manuscript. This is to confirm that no inappropriate, unethical or misleading image manipulation has occurred. These should be uploaded as 'Supporting information for review process only'. Please label/highlight the original gels so that we can clearly see which sections/lanes have been used in the manuscript figures. For more information, see: <https://physoc.onlinelibrary.wiley.com/hub/journal-policies#imagmanip>.

EDITOR COMMENTS

Reviewing Editor:

Ethics Concerns:
No concerns

Comments to the Author:

The revised manuscript by Estrada et al has shown that hypoinsulinemia in the cerebral spinal fluid may be a cause of the exaggerated exercise pressor reflex found in type 2 diabetics. The information contained in the manuscript is novel, interesting and potentially has important translational significance. Both reviewers have recommended acceptance of this manuscript, but one has questioned the quality of the western blots that are shown in figure 5. I share these concerns, and offer the authors a choice. They can either remove from their manuscript the data obtained from their western blots or they can redo their western blot figure so that it is acceptable. In part, one problem with figure 5E is that it shows th several of the blots merging across lanes; this is concerning because the plastic barriers creating lanes, are placed in the gel. As a consequence, the plastic barriers should not allow the blots to merge across lanes. Please explain how this could happen. There also appears to be substantial non-specific binding in the figure, making one question whether the authors optimized the concentration of the antibodies that they used. Please let me know what your decision is. Either way I will need to have your revised manuscript reviewed again by the same individuals.

Senior Editor:

Comments to the Author:

Thank you for submitting your revised manuscript. I have now received comments form the two original reviewers, both of whom recognise the importance of your work. However, as indicated by Reviewer 1, and emphasised by the Reviewing Editor, we are concerned about some of your Western blots. The Reviewing Editor has given you a choice, and I agree this is reasonable: either remove or redo your Western blots. We look forward to seeing your revision in due course.

REFEREE COMMENTS

Referee #1:

The authors did a good job of addressing most of my comments in the revised version. However, I still have concerns about the western blots.

Though images 5A, B, C, D, and F are not great I think they are okay. However, I don't think 5E is acceptable. The images have so much non-specific binding that it makes the interpretation of results questionable. Based on their methods (lines 335-337), and what is usually done, they stripped each blot after the primary antibody in order to do the housekeeping protein (beta-actin). I think all of those terribly thick additional bands are non-specific binding for IR-beta and this could be due to using a poor antibody, or too much of it, so it seems like they didn't optimize their antibodies first. And, this is likely why the remainder of the blots aren't very good either. There are many other causes of non-specific binding but I think antibody is the most likely.

I suggest that the authors add a vertical line on each of the western blot images between the MW marker band and the remainder of the blot to the right. This makes it clear to the reader that those lanes were cut and pasted over (lines 374 - 376 is where it is described).

This is minor, but under 'Data Handling,' the authors write that they cropped the three lowest MW markers (line 377) but the MW marker they are using has 11 bands and 9 are present in the images so they only cropped 2.

For Figure 6, the authors include p values for the graphs that are not significant but only * for those that are. I think they should add actual p values for those as well.

Referee #2:

No further comments for the authors.

END OF COMMENTS

1st Confidential Review

23-May-2024

Intracerebroventricular insulin injection acutely normalizes the augmented exercise pressor reflex in male rats with type 2 diabetes mellitus

Senior Editor:

Comments to the Author:

Thank you for submitting your revised manuscript. I have now received comments from the two original reviewers, both of whom recognize the importance of your work. However, as indicated by Reviewer 1, and emphasized by the Reviewing Editor, we are concerned about some of your Western blots. The Reviewing Editor has given you a choice, and I agree this is reasonable: either remove or redo your Western blots. We look forward to seeing your revision in due course.

Author Response: We appreciate the reasonable assessment of our Western blots by the reviewers. As requested, we have optimized and re-done the Western blot for insulin receptor β subunit (Fig. 5E) using what was left of our remaining samples.

Reviewing Editor:

Ethics Concerns:

No concerns

Comments to the Author:

The revised manuscript by Estrada et al has shown that hypoinsulinemia in the cerebral spinal fluid may be a cause of the exaggerated exercise pressor reflex found in type 2 diabetics. The information contained in the manuscript is novel, interesting and potentially has important translational significance. Both reviewers have recommended acceptance of this manuscript, but one has questioned the quality of the western blots that are shown in figure 5. I share these concerns, and offer the authors a choice. They can either remove from their manuscript the data obtained from their western blots or they can redo their western blot figure so that it is acceptable. In part, one problem with figure 5E is that it shows that several of the blots merging across lanes; this is concerning because the plastic barriers creating lanes, are placed in the gel. As a consequence, the plastic barriers should not allow the blots to merge across lanes. Please explain how this could happen. There also appears to be substantial non-specific binding in the figure, making one question whether the authors optimized the concentration of the antibodies that they used. Please let me know what your decision is. Either way I will need to have your revised manuscript reviewed again by the same individuals.

Author Response: We thank the reviewing editor for taking the time to carefully review our work. As requested, we have re-done the Western blot for insulin receptor β subunit (Fig. 5E) following some optimization of the conditions. As a result, Figure 5 now appears much improved. During our testing we observed that the original antibody dilutions give the clearest band signals. A detailed explanation of what was done is given below in the response to Referee #1. As for the merged signal between lanes in the original image, this can most likely be explained by overloading of secondary antibodies, as the total signal on the blot is derived

from incompletely stripped antibodies from previous rounds of immunodetection (Akt and β actin) plus the antibodies from the current round of immunodetection (insulin receptor β subunit). Because we had concerns about the amount of sample we had for experiments, this was the approach we needed to take. The text within the manuscript has also been updated accordingly (Lines 304, 315-316, and 527).

Referee #1:

The authors did a good job of addressing most of my comments in the revised version. However, I still have concerns about the western blots.

Though images 5A, B, C, D, and F are not great I think they are okay. However, I don't think 5E is acceptable. The images have so much non-specific binding that it makes the interpretation of results questionable. Based on their methods (lines 335-337), and what is usually done, they stripped each blot after the primary antibody in order to do the housekeeping protein (beta-actin). I think all of those terribly thick additional bands are non-specific binding for IR-beta and this could be due to using a poor antibody, or too much of it, so it seems like they didn't optimize their antibodies first. And, this is likely why the remainder of the blots aren't very good either. There are many other causes of non-specific binding but I think antibody is the most likely.

Author Response: As requested, we have re-done the Western blot for insulin receptor β subunit using what was left over of our remaining samples (Fig. 5E). As you can see, the thick heavy bands are no longer present in the image and the background signal is greatly reduced. Following some optimization tests, we think the original antibody dilutions produces the clearest image, as reduced dilutions result in images with low signal to noise ratio. Additionally, we tested several blocking reagents (milk, LICOR blocking reagent, Nacalai blocking reagent) and we determined that the strongest blocking buffer produces the clearest signals, while other blocking buffers result in increased non-specific binding and background noise. These were the two most important factors determining image quality. We also determined that reducing secondary antibody concentration had the least amount of influence on the signal, perhaps even diminishing the signal to noise ratio. Furthermore, though we sacrificed some sample to test protein loads of 2.5 to 20 ug, we chose 5 ug as this is reasonably close to the load used on the other blots and our remaining sample is also limited. After adjusting for loading with β actin, and then normalizing the densitometry data for the band of interest we observed no differences. Thus, we added this data to the original data set. We hope the reviewer finds this new image acceptable, as the previously seen thick bands are no longer present, and the band of interest is clearly visible. Please note that the imperfections within the first lane and the last lane are due to creases within the PVDF membrane formed during the manufacturing process. We could not fix this issue, even with extensive pre-soaking of the membrane in transfer buffer. The manuscript has been updated accordingly (Lines 304, 315-316, and 527).

I suggest that the authors add a vertical line on each of the western blot images between the MW marker band and the remainder of the blot to the right. This makes it clear to the reader that those lanes were cut and pasted over (lines 374 - 376 is where it is described).

Author Response: We have added a vertical line to make it clear to the reader that the protein ladder was cut and pasted into the image. We appreciate this suggestion as it adds visual clarity for the reader.

This is minor, but under 'Data Handling,' the authors write that they cropped the three lowest MW markers (line 377) but the MW marker they are using has 11 bands and 9 are present in the images so they only cropped 2.

Author Response: Thank you for bringing this mistake to our attention. We have made the corrections within the text of the manuscript (Line 377).

For Figure 6, the authors include p values for the graphs that are not significant but only * for those that are. I think they should add actual p values for those as well.

Author Response: We agree with the reviewer that for consistency within the graph we should add exact p-values for each comparison. The changes have been made (Fig.6). Thank you.

Referee #2:

No further comments for the authors.

Author Response: We kindly thank the reviewer for taking the time to review our manuscript.

Dear Dr Mizuno,

Re: JP-RP-2024-286715R2 "Intracerebroventricular insulin injection acutely normalizes the augmented exercise pressor reflex in male rats with type 2 diabetes mellitus" by Juan A Estrada, Rie Ishizawa, Han Kyul Kim, Ayumi Fukazawa, Amane Hori, Norio Hotta, Gary A Iwamoto, Scott A. Smith, Wanpen Vongpatanasin, and Masaki Mizuno

We are pleased to tell you that your paper has been accepted for publication in The Journal of Physiology.

Authors should note that it is too late at this point to offer corrections prior to proofing. Major corrections at proof stage, such as changes to figures, will be referred to the Editors for approval before they can be incorporated. Only minor changes, such as to style and consistency, should be made at proof stage. Changes that need to be made after proof stage will usually require a formal correction notice.

If you would like to receive our 'Research Roundup', a monthly newsletter highlighting the cutting-edge research published in The Physiological Society's family of journals (The Journal of Physiology, Experimental Physiology and Physiological Reports), please click this link, fill in your name and email address and select 'Research Roundup': <https://www.physoc.org/journals-and-media/membernews/>.

Yours sincerely,

Vaughan Macefield
Senior Editor
The Journal of Physiology

P.S. - You can help your research get the attention it deserves! Check out Wiley's free Promotion Guide for best-practice recommendations for promoting your work at www.wileyauthors.com/eoo/guide. You can learn more about Wiley Editing Services which offers professional video, design, and writing services to create shareable video abstracts, infographics, conference posters, lay summaries, and research news stories for your research at www.wileyauthors.com/eoo/promotion.

IMPORTANT NOTICE ABOUT OPEN ACCESS: To assist authors whose funding agencies mandate public access to published research findings sooner than 12 months after publication, The Journal of Physiology allows authors to pay an Open Access (OA) fee to have their papers made freely available immediately on publication.

You can check if your funder or institution has a Wiley Open Access Account here: <https://authorservices.wiley.com/author-resources/Journal-Authors/licensing-and-open-access/open-access/author-compliance-tool.html>.

EDITOR COMMENTS

Reviewing Editor:

Thank you for improving the quality of the western blot shown in figure 5.

Senior Editor:

Thank you of addressing the remaining comments by Reviewer 1. Both the reviewer and Reviewing Editor are satisfied with your changes and so I am pleased to report that your manuscript is now considered acceptable for publication in The Journal of Physiology.

REFEREE COMMENTS

Referee #1:

The authors provided adequate responses to my previous comments. The western blots in figure 5 still appear messy, but the explanation provided by the authors is clear and transparent. Therefore, I have no further comments.

2nd Confidential Review

02-Jul-2024